# Dual recognition of multiple signals in bacterial outer membrane proteins enhances assembly and maintains membrane integrity

Edward M Germany[1,2], Nakajohn Thewasano[1,2], Kenichiro Imai[3], Yuki Maruno[1,2], Rebecca S Bamert[4,5], Christopher J Stubenrauch[4,5], Rhys A Dunstan[4,5], Yue Ding[6,7], Yukari Nakajima[1,2], XiangFeng Lai[6], Chaille T Webb[4,5], Kentaro Hidaka[1,2], Kher Shing Tan[4,5], Hsinhui Shen[6,7]*, Trevor Lithgow[4,5]*, Takuya Shiota[1,2]*

[1]Frontier Science Research Center, University of Miyazaki, Miyazaki, Japan; [2]Organization for Promotion of Tenure Track, University of Miyazaki, Miyazaki, Japan; [3]Cellular and Molecular Biotechnology Research Institute, National Institute of Advanced Industrial Science and Technology (AIST), Tokyo, Japan; [4]Centre to Impact AMR, Monash University, Clayton, Australia; [5]Infection Program, Biomedicine Discovery Institute, and Department of Microbiology, Monash University, Clayton, Australia; [6]Department of Materials Science and Engineering, Monash University, Clayton, Australia; [7]Biomedicine Discovery Institute and Department of Biochemistry and Molecular Biology, Monash University, Clayton, Australia

*For correspondence:
hsin-hui.shen@monash.edu (HS);
trevor.lithgow@monash.edu (TL);
takuya_shiota@med.miyazaki-u.ac.jp (TS)

Competing interest: The authors declare that no competing interests exist.

**Abstract** Outer membrane proteins (OMPs) are essential components of the outer membrane of Gram-negative bacteria. In terms of protein targeting and assembly, the current dogma holds that a 'β-signal' imprinted in the final β-strand of the OMP engages the β-barrel assembly machinery (BAM) complex to initiate membrane insertion and assembly of the OMP into the outer membrane. Here, we revealed an additional rule that signals equivalent to the β-signal are repeated in other, internal β-strands within bacterial OMPs, by peptidomimetic and mutational analysis. The internal signal is needed to promote the efficiency of the assembly reaction of these OMPs. BamD, an essential subunit of the BAM complex, recognizes the internal signal and the β-signal, arranging several β-strands and partial folding for rapid OMP assembly. The internal signal-BamD ordering system is not essential for bacterial viability but is necessary to retain the integrity of the outer membrane against antibiotics and other environmental insults.

## eLife assessment

This **important** study reports the identification of a new amino acid sequence motif (i.e., "internal beta-signal") on outer membrane proteins, which is recognized by beta-assembly machinery in gram-negative bacteria. The authors carried out rigorous experiments, providing **compelling** evidence in support of their conclusions. This work significantly advances our understanding of the biogenesis of outer membrane proteins.

## Introduction

In Gram-negative bacteria, outer membrane proteins (OMPs) play important roles in membrane-mediated processes (*Lundquist et al., 2021*; *Tommassen, 2010*; *Walther et al., 2009*). Structurally, each major OMP has a β-barrel transmembrane domain, spanning the membrane as a cylindrical structure formed from amphiphilic, antiparallel β-strands (*Konovalova et al., 2017*; *Schulz, 2000*). Starting as nascent polypeptides synthesized by ribosomes in the cytoplasm, these newly synthesized OMPs cross the inner membrane in a process mediated by the SecYEG translocon (*Manting et al., 2000*; *Mori and Ito, 2001*), and then traverse the periplasm supported by the periplasmic chaperones (*Wang et al., 2021*). The molecular machine driving OMP insertion and assembly into the membrane is the β-barrel assembly machinery complex (BAM complex) (*Bakelar et al., 2016*; *Gu et al., 2016*; *Iadanza et al., 2016*). Given this essential role, and its location on the bacterial cell surface, the BAM complex has emerged as an attractive antibiotic target (*Hart et al., 2019*; *Imai et al., 2019*; *Kaur et al., 2021*; *Luther et al., 2019*).

In *Escherichia coli*, the BAM complex is composed of five subunits BamA, BamB, BamC, BamD, and BamE (*Bakelar et al., 2016*; *Gu et al., 2016*; *Iadanza et al., 2016*; *Wu et al., 2005*). BamA, the essential core subunit, is embedded in the outer membrane and projects an extensive N-terminal domain consisting of five polypeptide transport-associated (POTRA) repeats into the periplasm which interacts with other subunits (*Noinaj et al., 2013*; *Voulhoux et al., 2003*). There is some variation in the lipoprotein subunits that are present in the BAM complex varies in different bacterial lineages (*Anwari et al., 2012*; *Webb et al., 2012*) but both BamA and the lipoprotein BamD are found in the BAM complexes of all bacterial lineages and are essential for viability in *E. coli* (*Malinverni et al., 2006*; *Webb et al., 2012*; *Wu et al., 2005*). The interaction of the POTRA domains of BamA with its partner BamD contributes to a large, funnel-like feature of the BAM complex which expands into the periplasm.

BamA belongs to Omp85-superfamily protein conserved across bacterial lineages and organelles (*Heinz and Lithgow, 2014*; *Wu et al., 2005*). Omp85-superfamily proteins are themselves β-barrels, utilizing a transiently open lateral gate between the first and last β-strands as the catalytic site of the OMPs assembly (*Doyle et al., 2022*; *Doyle and Bernstein, 2021*; *Doyle and Bernstein, 2019*; *Noinaj et al., 2013*; *Takeda et al., 2021*; *Tomasek et al., 2020*; *Xiao et al., 2021*). Biophysical studies have shown 'snap-shot' structures of assembly intermediates of the BAM complex with substrate OMPs and demonstrated that during assembly the first strand of the lateral gate of the BamA interacts with the C-terminal strand of the substrate OMPs (*Doyle et al., 2022*; *Shen et al., 2023*; *Takeda et al., 2023*; *Tomasek et al., 2020*; *Wu et al., 2021*). OMPs contain a conserved sequence termed the 'β-signal' located within this C-terminus strand (*Kutik et al., 2008*; *Paramasivam et al., 2012*; *Struyvé et al., 1991*; *Wang et al., 2021*). Inhibitory compounds such as darobactin show a similar structure to that of the β-signal and are capable of interacting with the lateral gate (*Imai et al., 2019*; *Kaur et al., 2021*; *Xiao et al., 2021*). The current dogma suggested by these studies dictate that this β-signal should engage directly with the lateral gate in BamA (*Horne and Radford, 2022*; *Tomasek and Kahne, 2021*).

Despite recent advances in resolving the BAM complex structure and its function, the molecular mechanism of early-stage OMP assembly remains enigmatic. It has been shown that the β-signal plays a significant role in the targeting and folding of BAM complex substrates (*Kutik et al., 2008*; *Wang et al., 2021*), yet several important OMPs do not have a recognizable β-signal motif in their C-terminal sequence (*Celik et al., 2012*; *Hagan et al., 2015*). For instance, BamA itself lacks an apparent C-terminal β-signal (*Hagan et al., 2015*). Also, a comprehensive study of the autotransporter class of OMPs, including proteins such as EspP, showed that while 1038 non-redundant proteins did have a C-terminal β-signal motif, over 40% of these proteins had a β-signal motif only in internal strands (*Celik et al., 2012*), which was also suggested to be the case for BamA (*Hagan et al., 2015*). It remains to be tested if these internal motifs function as β-signals and how they engage the BAM complex.

Before substrate OMPs can enter the lateral gate of BamA, they must first pass through the periplasmic features of the BAM complex. Although the assembly reactions that occur at the lateral gate are well understood, less is known about the molecular mechanisms at the early stage, prior to being position in the lateral gate. The specific points that have been established are that the substrate makes contact with the BamB and BamD subunits and the lateral gate of the BamA subunit (*Hagan et al., 2013*; *Harrison, 1996*; *Ieva et al., 2011*).

Here, we establish a screening system based on an in vitro reconstituted membrane assay using a translocation-competent *E. coli* microsomal membrane (EMM) (*Gunasinghe et al., 2018*; *Thewasano et al., 2023*) and demonstrate that OMPs - including the major porins that are the predominant substrate of the BAM complex - contain multiple β-signal-related repeats. Comparison of the C-terminal β-signal with what we refer to here as the 'internal signal' revealed the essential elements of sequence and orientation in the signals that are recognized by the BAM complex. Internal signals and β-signals are recognized by BamD and are responsible for rapid assembly of the OMP into the bacterial membrane at the early stage. Moreover, in situ cross-linking and mutagenesis demonstrated that the specific relationship between the BAM complex and this dual signal plays an important role in the integrity of the outer membrane.

## Results

### Peptidomimetics derived from *E. coli* OmpC inhibit OMP assembly

The EMM assembly assay provides a means for in vitro reconstitution of BAM complex function, where the assembly of a $^{35}$S-labeled OMP can be monitored (*Figure 1—figure supplement 1A*; *Gunasinghe et al., 2018*; *Thewasano et al., 2023*). EspP, a model OMP substrate, belongs to autotransporter family of proteins. Autotransporters have two domains: (1) a β-barrel domain, assembled into the outer membrane via the BAM complex, and (2) a passenger domain, which traverses the outer membrane via the lumen of the β-barrel domain itself and is subsequently cleaved by the correctly assembled β-barrel domain (*Celik et al., 2012*). When EspP is correctly assembled into outer membrane, a visible decrease in the molecular mass of the protein is observed due to the self-proteolysis. Once the barrel domain is assembled into the membrane it becomes protease-resistant, with residual unassembled and passenger domains degraded (*Leyton et al., 2014*; *Roman-Hernandez et al., 2014*). When applied to $^{35}$S-labeled EspP, the assay discriminates assembly of a protease-protected β-barrel domain, assembled into the outer membrane by the BAM complex (*Ieva et al., 2011*), from the protease-sensitive extracellular passenger domain (*Figure 1A*) that are degraded by protease (*Figure 1—figure supplement 1B*). The addition of peptides with BAM complex affinity, such as the OMP β-signal, are capable of exerting an inhibitory effect by competing for binding of substrate OMPs to the BAM complex (*Hagan et al., 2015*). Thus, the addition of peptides derived from the entirety of OMPs to the EMM assembly assay, which can evaluate assembly efficiency with high accuracy, expects to identify novel regions that have affinity for the BAM complex. To screen for potential peptidomimetics that could inhibit BAM complex function, a synthetic peptide library was designed for coverage of the sequence features in OmpC (*Figure 1—figure supplement 2*). We used peptides that mapped the entirety of OmpC, with a two amino acid overlap. This we considered preferable to peptides that were restricted by structural features, such as β-strands, in consideration that β-strand formation may or may not have occurred in early-stage interactions at the BAM complex.

Six peptides (4, 10, 17, 18, 21, and 23) were found to inhibit EspP assembly (*Figure 1A*). Of these, peptide 23 corresponds to the canonical β-signal of OMPs: it is the final β-strand of OmpC and it contains the consensus motif of the β-signal ($\zeta$ xGxx[Ω/Φ]x[Ω/Φ]). The inhibition seen with peptide 23 indicated that our peptidomimetics screening system using EspP can detect signals recognized by the BAM complex. In addition to inhibiting EspP assembly, five of the most potent peptides (4, 17, 18, 21, and 23) inhibited additional model OMPs; the porins OmpC and OmpF, the peptidoglycan-binding OmpA, and the maltoporin LamB (*Figure 1—figure supplement 3*). Comparing the sequences of these inhibitory peptides suggested the presence of a sub-motif from within the β-signal, namely [Ω/Φ]x[Ω/Φ] (*Figure 1B*). The sequence codes refer to conserved residues such that: $\zeta$, is any polar residue; G is a glycine residue; $\Omega$ is any aromatic residue; $\Phi$ is any hydrophobic residue, and x is any residue (*Hagan et al., 2015*; *Kutik et al., 2008*). The non-inhibitory peptide 9 contained some elements of the β-signal but did not show inhibition of EspP assembly (*Figure 1A*).

Peptide 18 also showed a strong sequence similarity to the consensus motif of the β-signal (*Figure 1B*) and, like peptide 23, had a strong inhibitory action on EspP assembly (*Figure 1A*). Variant peptides based on the peptide 18 sequence were constructed and tested in the EMM assembly assay (*Figure 1C*). This analysis revealed that the position 0 (0Φ), in addition to the [Ω/Φ]x[Ω/Φ], motif was functionally important to inhibitory action (*Figure 1C*). Thus, we hypothesized the elements of an internal β-signal (hereafter the internal signal) to be $\Phi$xxxxx[Ω/Φ]x[Ω/Φ].

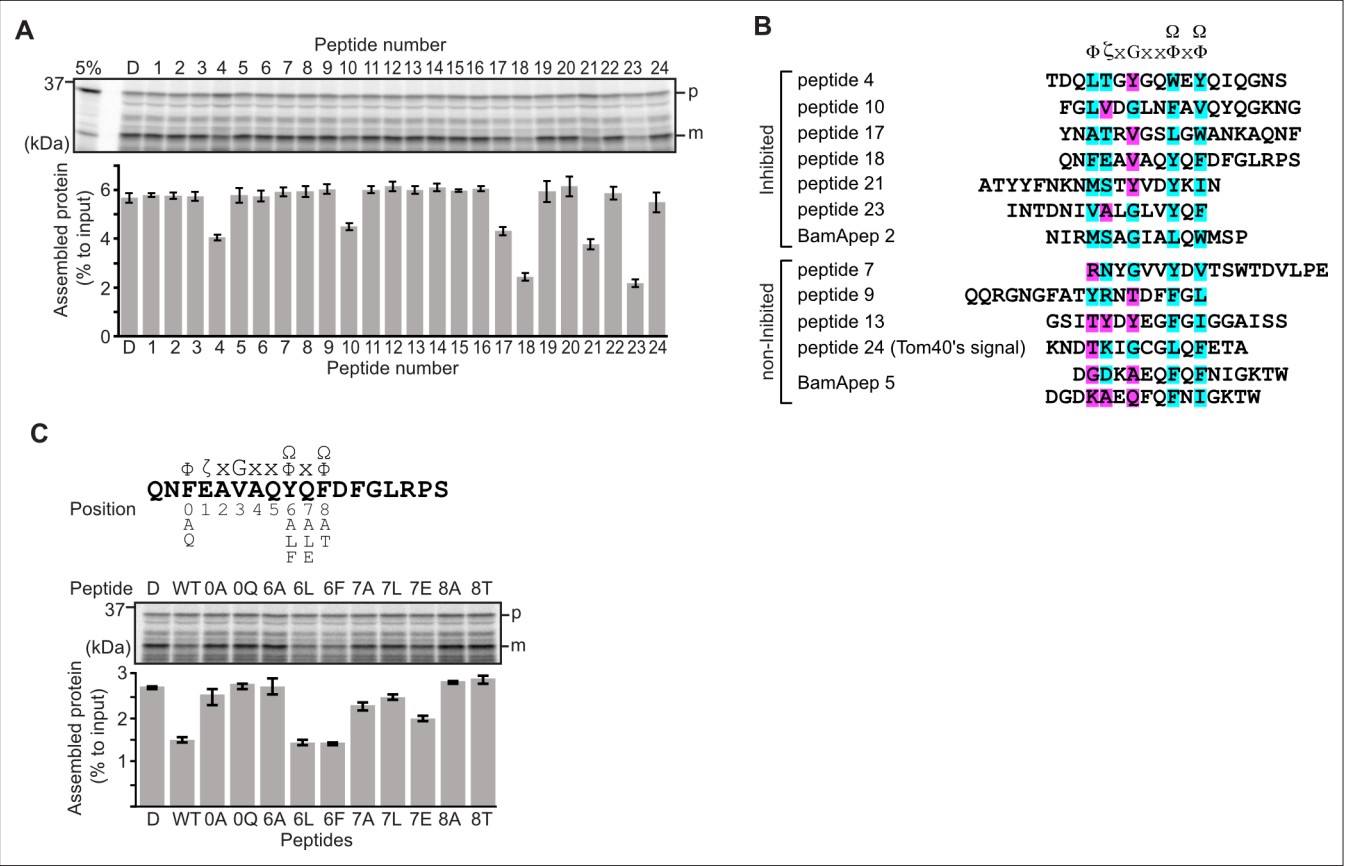

**Figure 1.** Peptides derived from β-strands of OmpC selectively inhibit outer membrane protein (OMP) assembly. (A and C) ³⁵S-labeled EspP was synthesized by rabbit reticulocyte lysate, then incubated with *E. coli* microsomal membrane (EMM), and samples prepared for analysis by SDS-PAGE and radio-imaging (see Materials and methods). EspP assembly into EMM was performed in the presence of DMSO (**D**) or specific inhibitor peptides. The precursor (**p**) and mature (**m**) form of EspP are indicated. (**A**) Peptides labeled 1 through 23 were derived from OmpC, while peptide 24 was taken from a mitochondrial β-barrel protein to serve as a negative control (see *Figure 1—figure supplement 2*). (**B**) Sequence comparison of inhibitor peptides identified in the EspP assembly assays. The signal comprises key hydrophobic (Φ), polar (ζ), glycine (**G**), and two aromatic (Ω) or hydrophobic amino acids. Amino acids highlighted in blue conform to the consensus signal, while those highlighted in purple deviate from the consensus signal. (**C**) Upper: sequence of the peptide 18 indicating residue codes for the mutated peptides, e.g., at position '6', the native residue Y was mutated to A in peptide '6A'. Position 0 designates the first hydrophobic residue and position 8 designates the final aromatic or hydrophobic residue. DMSO (**D**) was used for control. Middle: SDS-PAGE of EspP assembly assay. Lower: data for inhibition of EspP assembly into EMM assayed in the presence of each of the indicated peptides.

The online version of this article includes the following figure supplement(s) for figure 1:

**Figure supplement 1.** Schematic of *E. coli* microsomal membrane (EMM) isolation and EMM assembly assay.

**Figure supplement 2.** Design of peptide inhibitor library from the *E. coli* outer membrane protein (OMP) OmpC.

**Figure supplement 3.** Peptide inhibition of multiple β-barrel assembly machinery (BAM) complex substrate assembly.

## Mutations to a putative internal signal results in loss of OMP assembly

To assess the impact of the conserved residues in the potential internal signal for OMP assembly, a series of mutant OmpC variants were constructed and assayed for assembly in the EMM assay (*Figure 2A*). Three of the peptide regions are present in the final five β-strands of OmpC. We denoted these as the first (–1), third (–3), and fifth (–5) β-strands from the C-terminus of the barrel. Note, since peptide 17 and peptide 18 overlap, they were considered together as a single region. Given the antiparallel structure of a β-barrel the –1, –3, and –5 strands are orientated in the same direction, and these three strands all conform to the putative Φxxxxx[Ω/Φ]x[Ω/Φ] motif.

Blue native (BN)-PAGE analysis was employed to measure the efficiency of OmpC trimer assembly (*Figure 2A*). These gels separate membrane proteins that have been solubilized from native membranes, allowing time-course analysis of protein assembly. Among the mutants with significantly

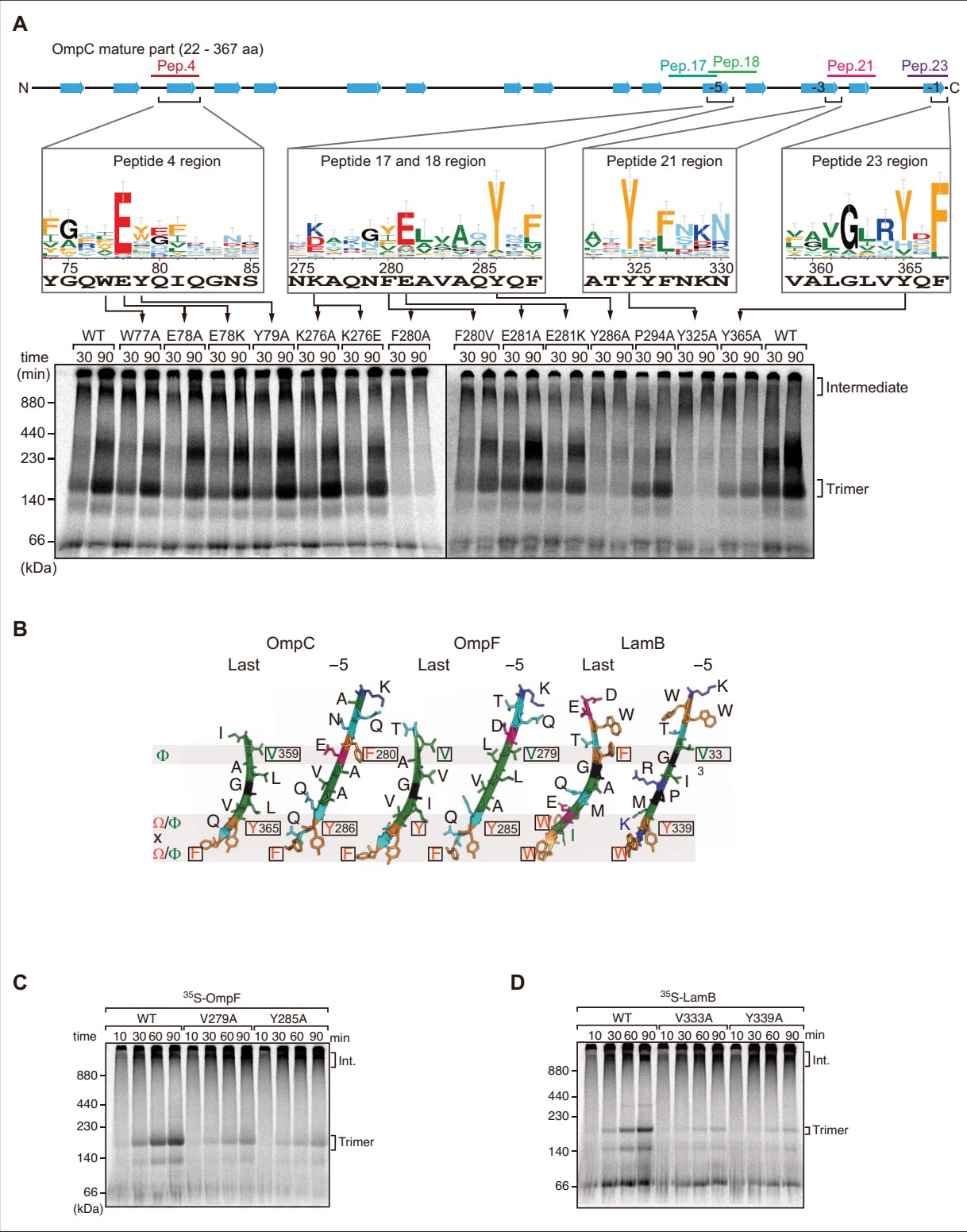

**Figure 2.** Mutations in β-signals inhibit outer membrane protein (OMP) assembly in vitro. (**A**) Representation of OmpC (upper panel) annotated to show the position of each β-strand (blue arrows) and the region mimicked in the indicated peptides. Sequence logos show the conservation of sequence for the residues in these regions (see Materials and methods), where the height of letter corresponds to the degree of conservation across bacterial species, residues are color-coded: aromatic (orange), hydrophobic (green), basic (blue), acidic (red), and proline and glycine (black). In the lower panel, data from the *E. coli* microsomal membrane (EMM) assembly assay of OmpC mutants is organized to show effects for key mutants: [35]S-labeled OmpC wild-type or mutant proteins were incubated with EMM for 30 or 90 min, and analyzed by Blue native (BN)-PAGE and radio-imaging.

*Figure 2 continued on next page*

*Figure 2 continued*

(**B**) Comparison of amino acids in the final and fifth to final (–5) β-strands of OmpC, OmpF, and LamB. All three OMPs contain the putative hydrophobic residue near the N-terminal region of the β-strands, highlight with a black box. A hydrophobic and an aromatic residue is also found to be highly conserved between these OMP β-strands at the C-terminal amino acids of the β-strands. (**C–D**) Assembly of ³⁵S-labeled OmpF (**C**) or LamB (**D**) with mutations to the –5 β-strand or final β-strand into EMM. Samples were prepared as in (**A**).

The online version of this article includes the following figure supplement(s) for figure 2:

**Figure supplement 1.** Mutations to internal β-signal-like motif results in loss of assembly in trimeric β-barrel assembly machinery (BAM) complex substrate, OmpC.

**Figure supplement 2.** Multiple alignment of –5 β-strand of outer membrane proteins (OMPs) with 8, 10, or 12 β-strands.

**Figure supplement 3.** Multiple alignment of –5 β-strand of outer membrane proteins (OMPs) with 14, 16, or 18 β-strands.

**Figure supplement 4.** Multiple alignment of –5 β-strand of outer membrane proteins (OMPs) with 22 or 24 β-strands.

reduced assembly efficiency, Y286A, Y325A, and Y365A correspond to Ω/Φ in position 6 (Ω/Φ), and F280A corresponds to Φ in position 0 of the putative signal (*Figure 2A* and *Figure 2—figure supplement 4*). In terms of the highly conserved residues present, sequence conservation was mapped against the β-strands of OMPs whose structures have been determined (*Figure 2*). Analysis showed the –5 strand of OmpF and LamB includes residues that conform to the putative Φxxxxx[Ω/Φ]x[Ω/Φ] motif (*Figure 2B*). We constructed mutants to the 0Φ or 6[Ω/Φ] positions of the –5 strand in OmpF and LamB, these are the OmpF(V279A), OmpF(Y285A), LamB(V333A), and LamB(Y339A) mutants (*Figure 2B*). BN-PAGE analysis showed that both the Φ – OmpF(V279) and LamB(V333) - and Ω/Φ – OmpF(Y285) and LamB(Y339) - residues were important for the assembly of OmpF and LamB (*Figure 2—figure supplement 4*).

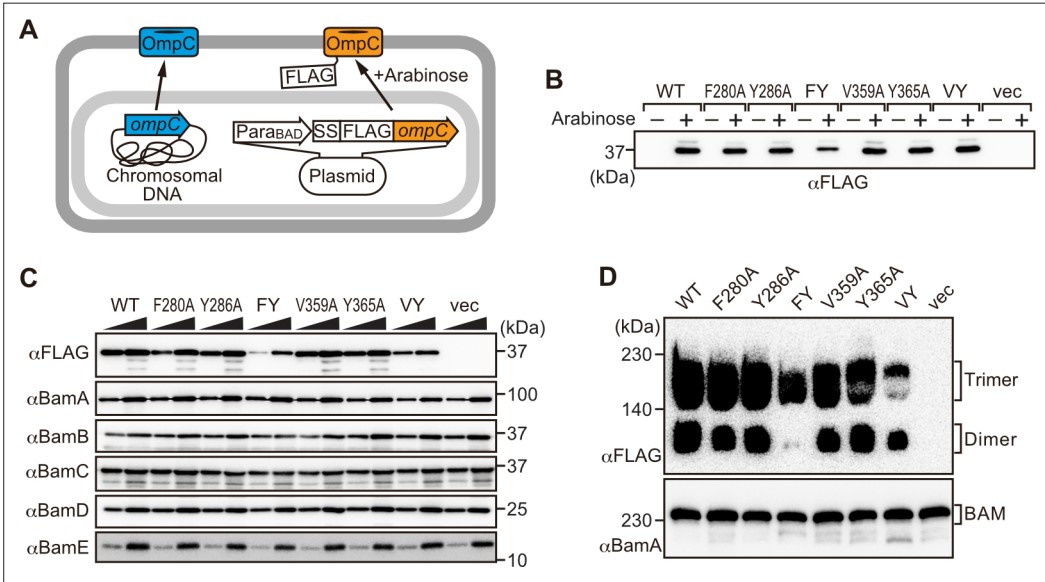

**Figure 3.** Modifications in the putative internal signal slow the rate of outer membrane protein (OMP) assembly in vivo. (**A**) Schematic model of FLAG-OmpC expression system in *E. coli*. The plasmid-borne copy of OmpC was mutated and the mutant proteins are selectively detected with antibodies to the FLAG epitope. (**B**) Total cell lysates were prepared from the variant *E. coli* strains expressing FLAG-OmpC(WT), –5 strand mutants: F280A, Y286A, double mutant F280A/Y286A (FY), or final β-strand mutants: V359A, Y365A, double mutant V359A/Y365A (VY) with (+) or without (-) arabinose. The steady-state levels of the indicated proteins were assessed after SDS-PAGE by immunoblotting with anti-FLAG antibody. (**C**) After growth with arabinose induction, *E. coli* microsomal membrane (EMM) fraction was isolated from each of the FLAG-OmpC variant expressing *E. coli* strains to assess steady-state levels of OmpC variants and β-barrel assembly machinery (BAM) complex subunits: BamA, BamB, BamC, BamD, and BamE by SDS-PAGE and immunoblotting. Triangles indicate the total protein amount of EMM in each sample: Left; 5 µg, right; 15 µg. (**D**) EMM fractions were solubilized with 1.5% DDM and subjected to Blue native (BN)-PAGE and immunoblotting. The indicated proteins were detected by anti-FLAG (top) and anti-BamA antibodies (bottom), respectively.

To address whether the results seen in vitro in the EMM assays were consistent with phenotypes drawn from intact *E. coli* cells, we constructed arabinose-inducible plasmids to express mutants of Φ (position 0), the [Ω/Φ] (position 6), and a double mutant combination of both positions in either the –5 or –1 strands of OmpC. To distinguish these plasmid-encoded OmpC variants from the endogenous, chromosome-encoded OmpC, a FLAG epitope tag was incorporated into the region of OmpC immediately behind the cleavage site for the secretory signal sequence (*Figure 3A*; *Rapoport, 2007*). Expression of FLAG-OmpC variants was induced with 0.1% arabinose, as confirmed through immunoblotting of total cell lysate (*Figure 3B*). In comparing the *E. coli* strains, a marked reduction was seen in the steady-state level of the double ('FY' or 'VY') mutants OmpC (F280A+Y286A or V359A+Y365A) (*Figure 3C*). Little change was evident in the expression levels of the single –1 mutants, suggesting that a fully conserved β-signal located at the C-terminus alone is not essential to maintaining OmpC assembly. To further investigate the nature of the defects, the various strains were analyzed by BN-PAGE. In all cases, OmpC that has been detergent-solubilized from outer membranes migrates as a characteristic population of dimeric and trimeric forms under conditions of the BN-PAGE (*Figure 3D*). Both the trimeric and dimeric forms of OmpC solubilized from the β-barrel arrays in the outer membrane have equivalent structural and functional properties (*Rocque and McGroarty, 1989*). The double ('FY' or 'VY') mutants displayed drastic reductions of the dimer and trimer (*Figure 3D*). As a control, the level and integrity of the BAM complex were monitored and found to be equivalent in all strains (*Figure 3C and D*). Taken together these results suggest either the internal signal (in strand –5) or the C-terminal β-signal (in strand –1) must be present to achieve efficient assembly of OmpC into the outer membrane.

## The internal signal is necessary for insertion step of assembly into OM

The β-signal located in the C-terminal β-strand is important for recognition by the BAM complex. In addition, because it is the last β-strand, it also contributes to engagement with the first β-strand in order that purified OMPs can fold into β-barrels, even in the absence of the BAM complex in artificial non-membrane environments such as detergent micelles (*Burgess et al., 2008*). To compare this property of strand engagement for β-barrel folding in detergent micelles, we purified urea-denatured OmpC and the OmpC variant proteins with mutations in the –5 strand (F280A and Y286A) and the –1 strand (V359A and Y365A). The spontaneous folding of each OmpC variant was assessed by the rate of formation of trimer in the presence of detergent micelles (*Figure 4—figure supplement 1A*). Trimer embedded into detergent micelles migrated at 100 kDa in this assay system, whereas non-folded monomeric OmpC migrated at 37 kDa (*Figure 4—figure supplement 1*). Relative to the rate/extent of trimer formation observed for OmpC(WT), mutations to the [Ω/Φ] (position 6) in the –5 strand, i.e., OmpC(Y286A) or the –1 strand, i.e., OmpC(Y365A) showed significantly impaired spontaneous folding. Position 6 is an amino acid that contributes to the aromatic girdle. Since Y286A and Y365A affected OMP folding as measured in folding experiments, their positioning into the aromatic girdle may contribute to the efficiency of β-barrel folding, in addition to contributing to the internal signal. Mutation of the Φ residue (position 0) in the –5 strand, i.e., OmpC(F280A) or the –1 strand, i.e., OmpC(V359A) maintained a similar ability to assemble into micelles as native OmpC. Thus, the conserved features of both –1 and –5 strands have equivalent impact on intrinsic properties for spontaneous folding of β-barrel of OmpC.

We further analyzed the role of internal β-signal by the EMM assembly assay. Compared to the rate at which the trimeric form of OmpC is assembled (*Figure 4A*), OmpC(Y286A) is slow and the gels resolve that most of the [35S]-labeled OmpC(Y286A) is held in a high-molecular-weight species for most of the time-course. Only at 80 min of assembly has substantial amount of the [35S]-labeled OmpC(Y286A) been resolved from this assembly intermediate ('Int') to a native, trimeric form (*Figure 4A*). In analyzing assembly intermediates of other sorts of membrane proteins in mitochondria, a version of this analysis referred to as 'gel shift' BN-PAGE analysis has been developed (*Shiota et al., 2012*). Here, the addition of an antibody recognizing the surface-exposed BamC to the samples prior to electrophoresis resulted in a dramatic shift, i.e., retardation of electrophoretic mobility due to the added mass of antibodies. This is consistent with OmpC(Y286A) being held in an assembly intermediate engaged with the BAM complex (*Figure 4B*). Consistent with this, the assembly intermediate which was prominently observed at the OmpC(Y286A) can be extracted from the membranes with urea; urea extraction of the membrane samples prior to analysis by BN-PAGE showed that the integral membrane trimer form

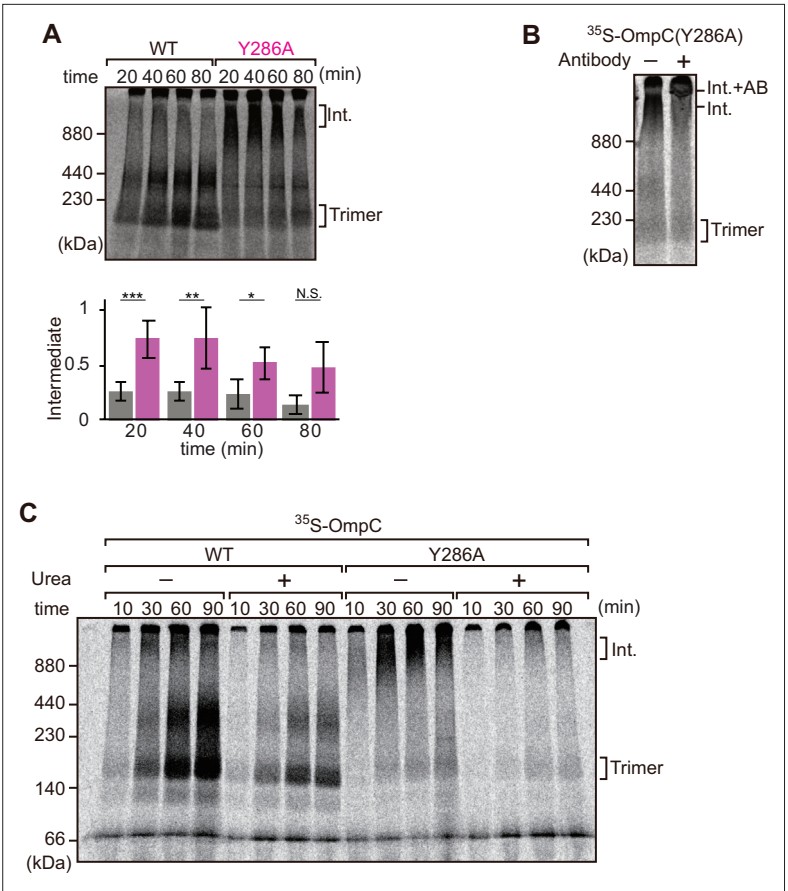

**Figure 4.** Mutation that stalls outer membrane protein (OMP) assembly in a high-molecular-weight assembly intermediate. (**A**) Upper panel, assembly assay of $^{35}$S-OmpC wild-type (WT) and mutant Y286A. After incubation in the *E. coli* microsomal membrane (EMM) assays for indicated time points, $^{35}$S-proteins were resolved via Blue native (BN)-PAGE. OmpC trimer and intermediate (int) are indicated. The lower panel shows a quantification of the intermediate form compared to total assembled protein (average of three independent experiments). Statistical significance was indicated by the following: N.S. (not significant), p<0.05; *, p<0.005; **, p<0.001; ***. Exact p values of intermediate formed by WT vs Y286A at each time point were as follows: 20 min: p=0.00101, 40 min: p=0.006, 60 min: p=0.0371, 80 min: p=0.1283. (**B**) $^{35}$S-labeled OmpC(Y286A) was incubated with EMM for 60 min, then the EMM fraction was solubilized with 1.5% DDM and incubated with (+) or without (-) anti-BamC antibody for 40 min. These samples were then analyzed by BN-PAGE. The presumptive translocation intermediate (Int) and the gel shift species (Int. +AB) are indicated. (**C**) $^{35}$S-labeled OmpC was incubated with EMM and then analyzed by BN-PAGE and radio-imaging. EMMs were then washed and solubilized in the presence (+) or absence (-) of 6 M urea, after which the membrane fraction was re-isolated via ultracentrifugation for 45 min. Isolated membranes were solubilized in 1.5% DDM and analyzed by BN-PAGE.

The online version of this article includes the following figure supplement(s) for figure 4:

**Figure supplement 1.** In vitro folding of OmpC β-signal mutants into detergent micelles.

---

of OmpC and OmpC(Y286A) was not extracted by urea (*Figure 4C*). Conversely, the assembly intermediate was completely extracted with urea, as expected for a substrate engaged in protein-protein interactions with a proteinaceous complex in the membrane fraction. These results suggest that the mutation in OmpC(Y286A) causes a longer interaction with the BAM complex, as a species of protein that is not yet inserted into the surrounding membrane environment.

Three subunits of the BAM complex have been previously shown to interact with the substrates: BamA, BamB, and BamD (*Hagan et al., 2013*; *Harrison, 1996*; *Ieva et al., 2011*). In vitro pull-down assay showed that while BamA and BamD can independently bind to the in vitro translated OmpC polypeptide (*Figure 5—figure supplement 1A*), BamB did not (*Figure 5—figure supplement 1B*). To test for signal-dependent binding, BamA and BamD were pre-incubated with inhibitor peptides

(*Figure 5A*). Peptide 23 inhibited binding of the substrate by BamA and peptide 18 inhibited binding of the substrate by BamD. Peptide 21 inhibited binding of the substrate by either BamA or BamD. These results suggest that BamA and BamD are important for recognition of both the β-signal and the internal signal. While the biochemical assay demonstrated that the OmpC(Y286A) mutant forms a stalled intermediate with the BAM complex, in a state in which membrane insertion was not completed, biochemical assays such as this cannot elucidate where on BamA-BamD this OmpC(Y286A) substrate is stalled. Neutron reflectometry (NR) is a powerful tool for probing the details of substrate-chaperone interactions, and is ideally suited for membrane-embedded proteins (*Shen et al., 2014*). The rationale to these experiments is that NR: (1) provides information on the distance of specified subunits of a protein complex away from the atomically flat gold surface to which the complex is attached, and (2) allows the addition of samples between measurements, so that multi-step changes can be made to, for example, detect changes in domain conformation in response to the addition of a substrate. To apply NR imaging to monitor the engagement of OmpC with the BAM complex, BamA was immobilized on an Ni-NTA atomic flat gold-coated silicon wafer to orient the POTRA domain distal to the silicon wafer as previously described (*Ding et al., 2020*). A lipid layer was reconstituted to provide BamA with a membrane-like environment prior to sequential measurement of: (1) BamA with lipid (first measurement), (2) BamA with lipid and BamD (second measurement), and (3) BamA-BamD-lipid with the addition of OmpC(WT) or OmpC(Y286A) (third measurement) (*Figure 5—figure supplement 2A*).

Comparing the reflectivity profiles of three measurements (*Figure 5B*), changes were observed in the layers after the addition of BamD and OmpC(Y286A), as seen in a shift of fringe to low Q range around 0.02–0.03 Å$^{-1}$. Crystal structures show that the periplasmic domains of BamA can be treated as two rigid bodies: POTRA1-POTRA2(P1-2) and POTRA3-POTRA4-POTRA5(P3-5). Thus, for further data analysis, we analyzed BamA as four layers: the His$_6$ extracellular layer where BamA is attached to the chip, the membrane layer (containing the β-barrel domain of BamA), an adjoining P3-5 layer, and the N-terminal P1-2 layer. NR data can then highlight variations in thickness of these layers, which corresponds to the position of additional protein mass in each point of measurement (*Figure 5C*, *Figure 5—figure supplement 2B, C*, and *Figure 5—source data 1*).

On addition of BamD (second measurement), the scattering length density (SLD) profiles showed that BamD was located in the P3-5 layer (*Figure 5C*, *Figure 5—figure supplement 2D, E*, and *Figure 5—source data 2*), an observation consistent with previous studies on the resting BAM complex (*Ding et al., 2020*). On addition of the OmpC(Y286A) substrate (third measurement) binding was observed to the P3-5 layer, with a concomitant extension of this layer from 29.7±0.8 Å to 45.5±1.6 Å away from the membrane (*Figure 5C and D*, *Figure 5—figure supplement 2F, G*, and *Figure 5—source data 3*). By contrast, no obvious changes were observed on the addition of the native OmpC protein implying that interactions it makes with BamA or BamD on the Si-Wafer were resolved too quickly to capture (*Figure 5—figure supplement 3*). The failure to observe changes with the WT-OmpC control in the NR analysis is consistent with the data from the BN-PAGE analysis (*Figure 4A*) where native OmpC does not form the stable intermediate seen for OmpC(Y286A). The NR findings indicate that OmpC(Y286A) is stacked at a position in the BAM complex with the POTRA3-5 of BamA and BamD, and that the Y286A mutation results in a relatively stable binding to this specific periplasmic region of the BAM complex.

## BamD acts as a receptor for the internal signal and β-signal

To elucidate the residues within BamD that mediate this signal-receptor interaction, we applied in vitro site-specific photo-cross-linking (*Figure 6—figure supplement 1A*). A library of 40 BamD variants was created to incorporate the non-natural amino acid, *p*-benzoyl-L-phenylalanine (BPA), by the suppressor tRNA method in a series of positions (*Chin et al., 2002*). We purified 40 different BPA variants of BamD, and then irradiated UV after incubating with $^{35}$S-labeled OmpC. The UV-dependent appearance of cross-links between OmpC and BamD occurred for 17 of the BamD variants (*Figure 6—figure supplement 1B*). Structurally, these amino acids locate both the lumen side of funnel-like structure (e.g. 49 or 62) and outside of funnel-like structure such as BamC-binding site (e.g. 114) (*Figure 6—figure supplement 1C*). The analysis was repeated to determine BamD cross-links for $^{35}$S-labeled OmpC(WT), the internal signal mutant OmpC(Y286A), or the β-signal mutant OmpC(βAAA: Y365A, Q366A, F367A) (*Figure 6—figure supplement 1D*). This mapping of OmpC

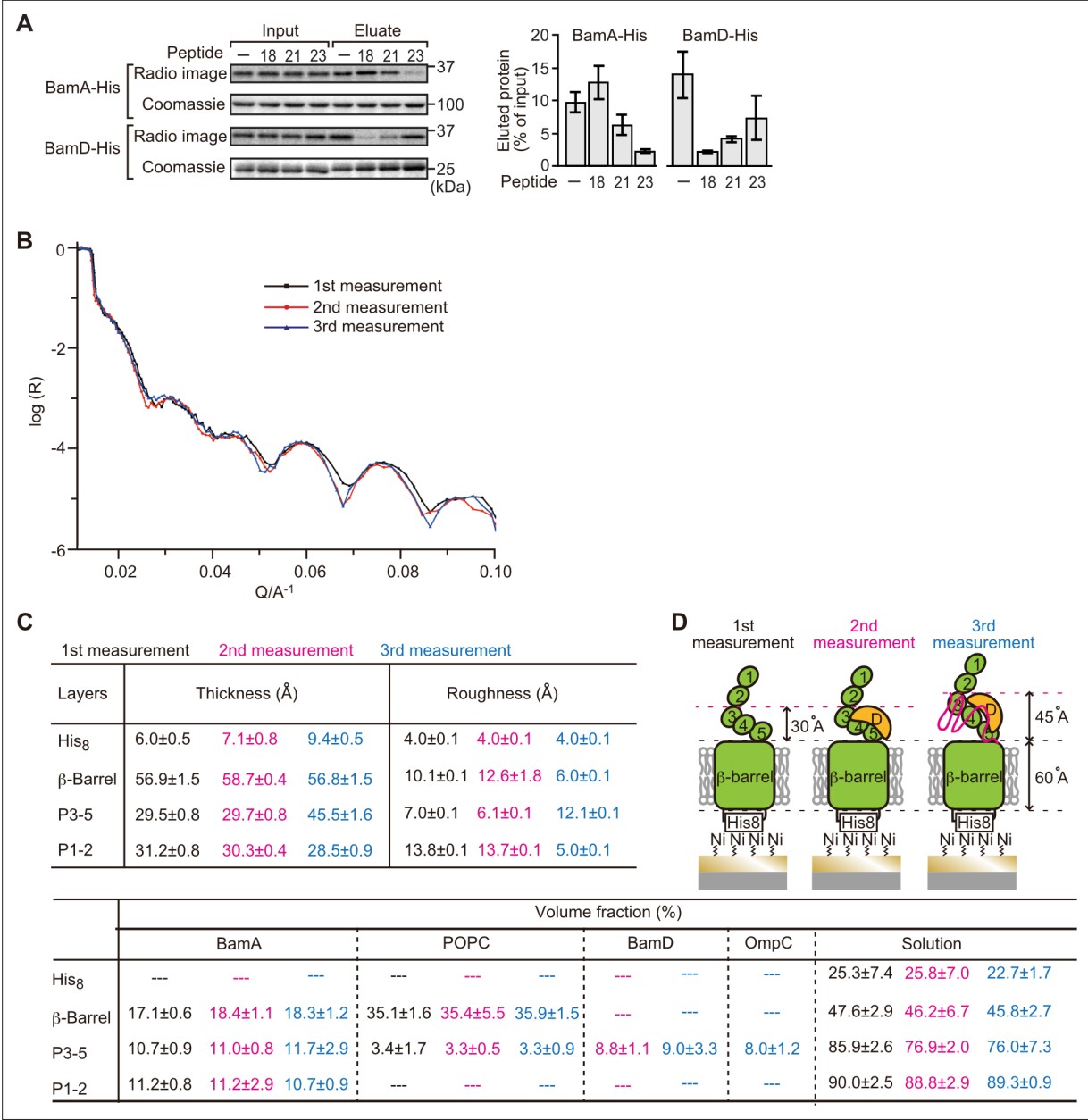

**Figure 5.** Proximity and dependence of BamD with OmpC(Y286A) revealed by neutron reflectometry (NR) analysis. (**A**) Ni-NTA was used to affinity purify His6-BamA or His6-BamD. The protein-Ni-NTA bead complex was then incubated in the presence of DMSO or the indicated peptides resuspended in DMSO, and $^{35}S$-labeled OmpC was incubated with these beads. After washing, bound proteins were eluted with 400 mM imidazole buffer. The graph represents densitometry analysis of data from three independent experiments. (**B**) NR profiles for BamA in membrane (first measurement, square black line), after addition of BamD (second measurement, circle red line), and after addition of the OmpC(Y286A) substrate (third measurement, triangle blue line). The experimental data was fitted using a seven-layer model: chromium - gold - NTA - His8 - β-barrel - P3-5 - P1-2. (**C**) Table summaries of the thickness, roughness, and volume fraction data of each layer from the NR analysis. The thickness refers to the depth of layered structures being studied as measured in Å. The roughness refers to the irregularities in the surface of the layered structures being studied as measured in Å. First, second, and third measurement displayed in black, magenta, and blue, respectively. P3-5: POTRA3, POTRA4, and POTRA5, P1-2: POTRA1 and POTRA2, lipid: POPC, solution: D2O, gold match water (GMW), and H2O. Details were described in **Supplementary file 8** to **Supplementary file 10**. (**D**) NR schematic of BamA (green), BamD (yellow), and OmpC(Y286A) (magenta). Numbers indicate corresponding POTRA domains of BamA. Note that the cartoon is a depiction of the results and not a scale drawing of the structures.

The online version of this article includes the following source data and figure supplement(s) for figure 5:

**Source data 1.** NR data of the first mesurement.

**Source data 2.** NR data of the second mesurement.

*Figure 5 continued on next page*

*Figure 5 continued*

**Source data 3.** NR data of the third mesurement.

**Figure supplement 1.** Ni-NTA pull-down of 35S-OmpC with purified β-barrel assembly machinery (BAM) complex substrate-binding subunits.

**Figure supplement 2.** Characterization of BamA-POPC, BamA-POPC BamD, and BamA-POPC-BamD-OmpC (Y286A) interaction via neutron reflectometry (NR).

**Figure supplement 3.** Characterization of BamA-POPC and BamA-POPC-BamD complex formation via neutron reflectometry (NR).

binding showed that an N-terminal section of BamD (amino acid residue 60 or 65) interacts with the internal signal, while a distinct section of BamD (amino acid residues 196, 200, 204) interacts with β-signal (*Figure 6—figure supplement 1E*).

While in vitro site-specific photo-cross-linking provides information on the substrate receptor region of BamD, this data only indirectly measures which residues of OmpC are being recognized. As our peptidomimetic screen identified conserved features in the internal signal, and cross-linking high-lighted the N-terminal and C-terminal TPR motifs of BamD as regions of interaction with OmpC, we focused on amino acids specifically within the β-signals of OmpC and regions of BamD which interact with β-signal. We purified a form of BamD that is substituted with a cysteine residue in the substrate-binding region, and synthesized forms of $^{35}$S-labeled OmpC substituted with a cysteine residue at various positions. Treatment with 1 µM $CuSO_4$ can stimulate disulfide formation between BamD and OmpC (*Figure 6A*). Consistent with the BPA cross-linking data, the positions near the internal signal in OmpC (amino acid residues 278, 284, 296, and 302) formed disulfide bonds with the N-terminal section of BamD (amino acid residues 49, 65). Conversely, residues toward the C-terminal, canonical β-signal of OmpC (amino acid residues 333, 357, and 363) formed disulfide bonds with a C-terminal section of BamD (at amino acid residue 204) (*Figure 6B*). All disulfide cross-linked products were validated by the addition of reducing condition of SDS-PAGE. (*Figure 6—figure supplement 2*).

To further probe this interaction, we engineered an *E. coli* strain to express BPA-containing BamD and a FLAG-tagged form of OmpC and performed in vivo photo-cross-linking (*Figure 6C*). These western blots reveal cross-linked products representing the interacting protein species. Photo-cross-linking of unnatural amino acid is not a 100% efficient process, so the level of cross-linked products is only a small proportion of the molecules interacting in the assays. In BamD, BPA at positions 49, 53, 65, or 196 generated cross-links to FLAG-OmpC. Positions 114 and 200 in BamD were not detected to interact with OmpC. Positions 49, 53, 65, and 196 of BamD face the interior of the funnel-like structure of the periplasmic domain of the BAM complex, while position 114 is located outside of the funnel-like structure (*Bakelar et al., 2016*; *Gu et al., 2016*; *Iadanza et al., 2016*). We note that while position 114 was cross-linked with OmpC in vitro using purified BamD, that this was not seen with in vivo cross-linking. Instead, in the context of the BAM complex, position 114 of BamD binds to the BamC subunit and would not be available for substrate binding in vivo (*Bakelar et al., 2016*; *Gu et al., 2016*; *Iadanza et al., 2016*).

## Two structurally distinct regions of BamD promote OmpC folding and assembly

In order to directly measure how the β-strands of OmpC come together, we established a new assay in which the folding of purified OmpC can be measured with or without assistance by purified BamD. Using the crystal structure of OmpC as a guide, cysteine residues were engineered in place of amino acids that were outward-facing on the final five β-strands of folded OmpC, placed close enough to a cysteine residue in the adjacent β-strand to allow for the possibility of disulfide formation once the strands had been correctly aligned in the folded protein (*Figure 7A*). These cysteine-variant OmpC proteins were incubated with or without BamD to determine how efficiently disulfide bonds formed. Samples harboring cysteine residues in OmpC that were incubated with BamD migrated faster than non-Cys OmpC on non-reducing SDS-PAGE (*Figure 7B*). This migration difference was not observed in reducing conditions, validating that the faster migration occurred as a result of internal molecule disulfide bond formation. Disulfide bonds were formed in each of the paired strands, β-strands from −1 to −5, suggesting that BamD directly stimulates arrangement of C-terminal five strands of BamD.

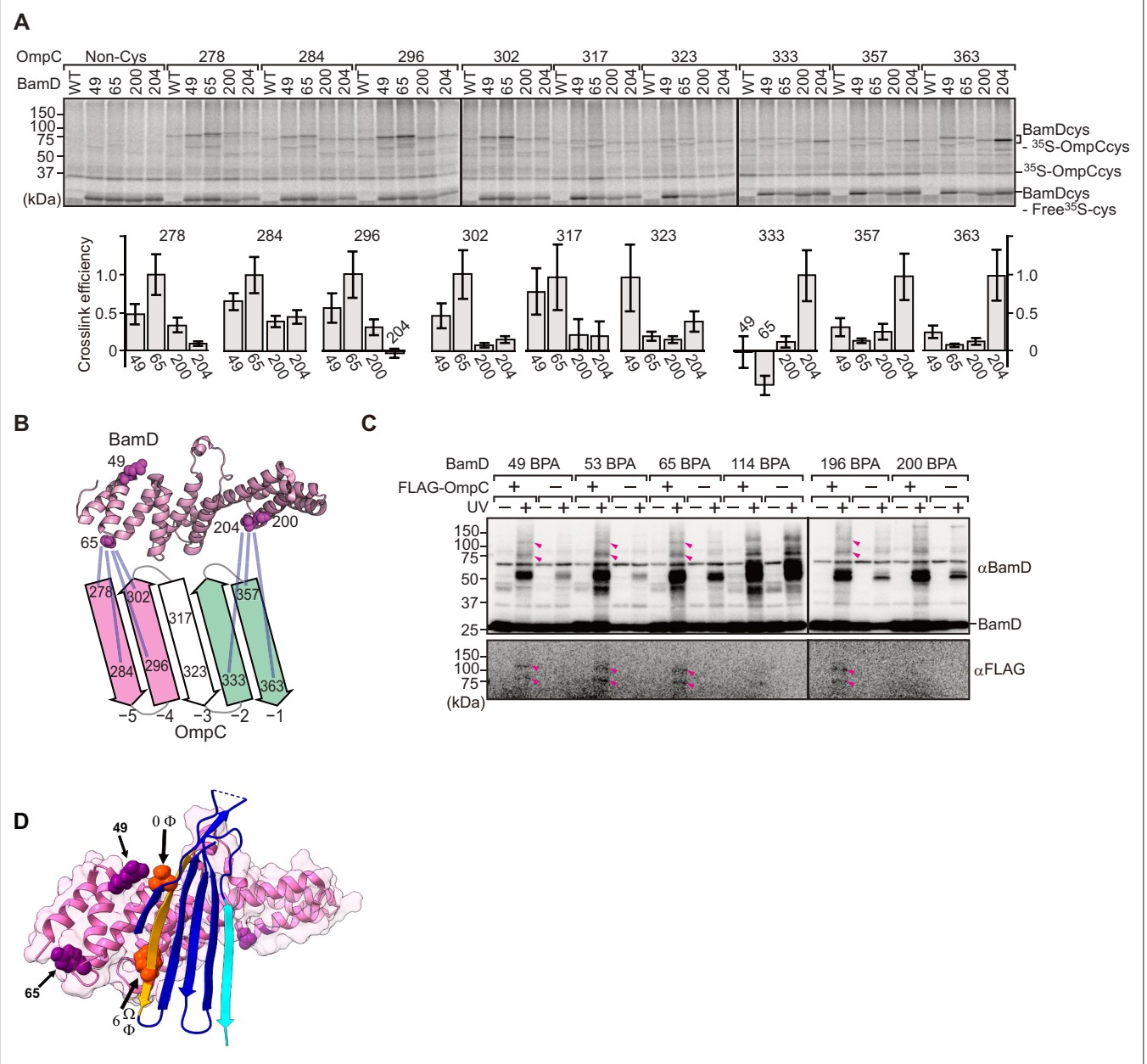

**Figure 6.** BamD acts as the receptor for internal β-signals. (**A**) BamD-cys mutant proteins were purified and incubated with ³⁵S-OmpC with cysteine mutations from the –5 to the final β-strand. After the addition of the oxidizing agent, CuSO₄, samples were purified with Ni-NTA and analyzed by SDS-PAGE. (**B**) BamD cysteine mutants, highlighted in magenta, formed disulfide bonds at distinct locations of OmpC. (**C**) p-Benzoyl-L-phenylalanine (BPA) containing BamD was expressed in the presence (+) or absence (-) of FLAG-OmpC overexpression. Pink arrows indicate BamD-FLAG-OmpC cross-link products. (**D**) Outer membrane protein (OMP) substrate superimposed on BamD. BamD cross-linking positions corresponding to internal signal are marked in purple and substrate cross-linking regions are marked in orange.

The online version of this article includes the following figure supplement(s) for figure 6:

**Figure supplement 1.** In vitro p-benzoyl-L-phenylalanine (BPA) cross-link analysis of substrate-binding regions within BamD.

**Figure supplement 2.** Analysis of the interaction between BamD and OmpC by intermolecular disulfide cross-linking.

Cross-linking studies shown in *Figure 6B* indicated two distinct regions of BamD as hotspots for OmpC binding. Anticipating that binding sites would be conserved, we analyzed sequence conservations and found BamD(Y62) and BamD(R197) as good candidates proximal to OmpC-binding sites (*Figure 7C*). R197 has previously been isolated as a suppressor mutation of a BamA temperature-sensitive strain (*Ricci et al., 2012*). Each residue was located on the lumen side of the funnel-like

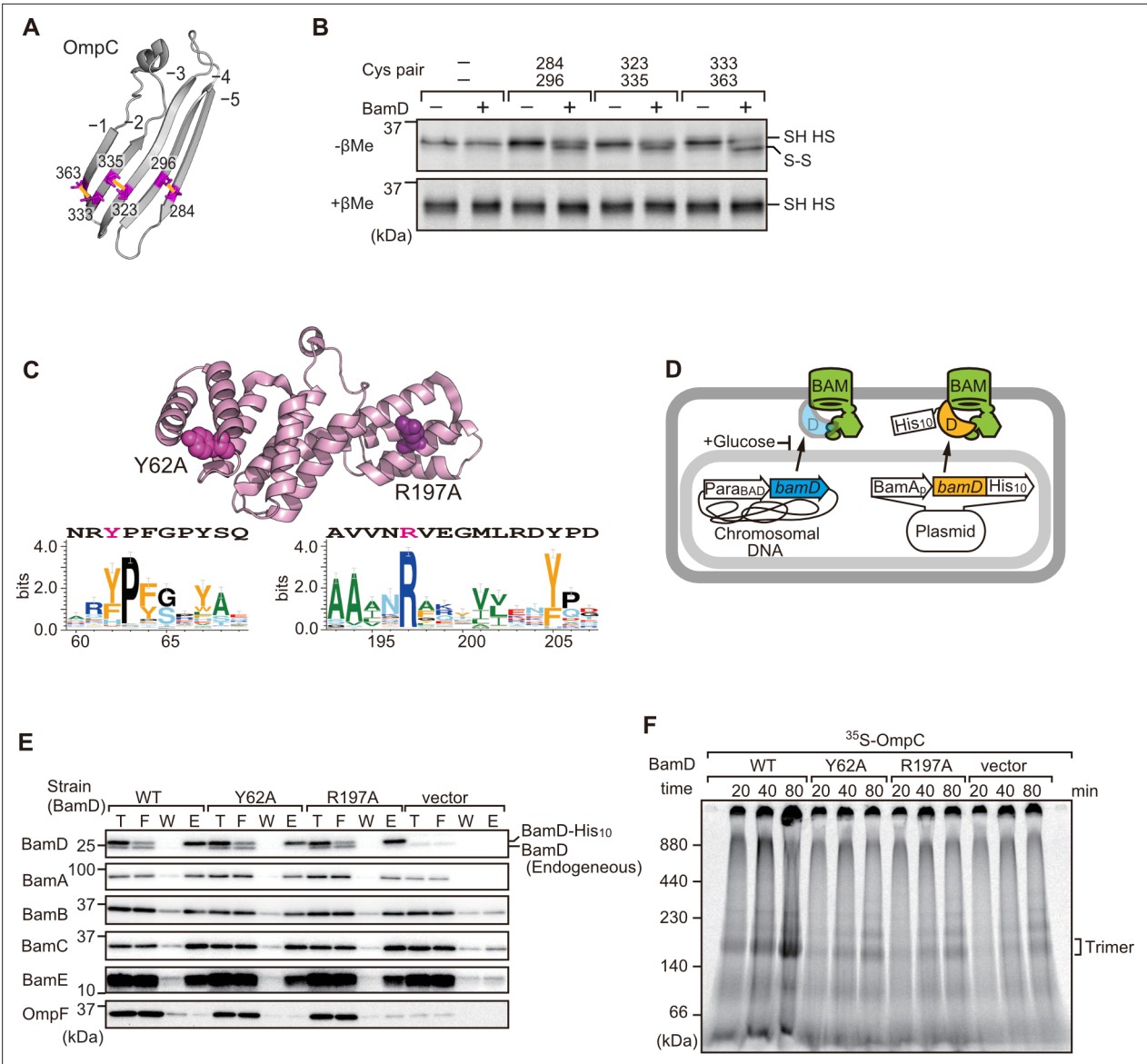

**Figure 7.** Key residues in two structurally distinct regions of BamD promote β-strand formation and outer membrane protein (OMP) assembly.
(**A**) Double cysteine residues were mutated into OmpC between anti-parallel β-strands to form artificial disulfide bonds. (**B**) 35S-OmpC-cys variants were translated then incubated with (+) or without (-) BamD. The samples were analyzed by reducing or non-reducing condition. Disulfide bond-specific bands (**S-S**) are indicated to the right of the gel. (**C**) Sequence conservation analysis of BamD. Target residues, Y62 and R197, are indicated with dark pink spheres on the crystal structure of BamD, sequence logos represent conservation in the immediate region of these residues. Amino acids listed above logo sequence are from crystal structure. Sequence conservation of BamD (bottom). (**D**) Schematic depiction of the BamD depletion strain of *E. coli* used to express mutant BamD proteins. (**E**) Pull-down assay of the variants of BamD-His8. The *E. coli* microsomal membrane (EMMs) as in (**C**) were solubilized with 1.5% DDM and then subjected to Ni-NTA. T - 5% total input, F - unbound fraction, W - wash fraction, E - eluted fraction. Each fraction was analyzed by SDS-PAGE and immunoblotting against indicated antibodies. (**F**) Assembly of OmpC was reduced in EMMs containing BamD mutated at Y62 and R197. 35S-labeled OmpC was incubated with EMM as in (**C**), and then analyzed by Blue native (BN)-PAGE and radio-imaging.

The online version of this article includes the following figure supplement(s) for figure 7:

**Figure supplement 1.** Validation and characterization of BamD depletion strain.

**Figure supplement 2.** Effect of BamD mutations on cell growth and β-barrel assembly machinery (BAM) complex formation in vivo and in vitro assembly ability.

**Figure supplement 3.** Schematic model of BamD depletion strain construction.

structure in the EspP-BAM assembly intermediate structure (PDBID; 7TTC) (*Doyle et al., 2022*). Mutant strains expressing BamD(Y62A) and BamD(R197A) in a genetic background where the chromosomal bamD can be depleted by an arabinose-responsive promoter were tested (*Figure 7D* and *Figure 7—figure supplement 1A–C*). Growth conditions were established wherein we could deplete endogenous BamD but not disturb cell growth (*Figure 7—figure supplement 1D*, and *Figure 7—figure supplement 2A*). Isolated EMM from this time point showed that neither the BamD(Y62A) nor the BamD(R197A) mutants affected the steady-state protein levels of other components of the BAM complex (*Figure 7—figure supplement 2B*), nor did they impact the complex formation of the BAM complex (*Figure 7E* and *Figure 7—figure supplement 2C*). The EMM assembly assay showed that the internal signal binding site was as important as the β-signal binding site to the overall assembly rates observed for OmpC (*Figure 7F*), OmpF (*Figure 7—figure supplement 2D*), and LamB (*Figure 7—figure supplement 2E*). These results suggest that recognition of both the C-terminal β-signal and the internal signal by BamD is important for efficient protein assembly.

## BamD is responsible for OMP assembly to retain outer membrane integrity

In *E. coli*, BamD function is essential for cell viability (*Onufryk et al., 2005*). As a result, in an *E. coli* strain where bamD expression is under the control of an arabinose-inducible promoter, in a control strain (*Figure 8A*, 'vec') two rounds of dilution in restrictive growth media deplete the level of BamD protein and stop cell growth (*Figure 8A*). The BamD(Y62A) mutant and the BamD(R197A) mutant supported viability (*Figure 8A* and *Figure 7—figure supplement 2A*), allowing membrane fractions to be isolated from each mutant strain: the steady-state level of porins OmpA and OmpC/OmpF was reduced in the BamD(Y62A) and BamD(R197A) mutants (*Figure 8B*). This was also true when the level of trimeric OmpF was assessed by native PAGE: less OmpF trimer had been assembled over the growth period relative to the steady-state level of the BAM complex (*Figure 8C*). Thus, while these mutant forms of BamD support cell viability (*Figure 8A*), they have deleterious effects on β-barrel protein assembly in vivo. In the case of the BamD(Y62A) strain, growth in the presence of vancomycin showed impaired outer membrane integrity (*Figure 8D and E*).

Vancomycin sensitivity is a classic assay to measure membrane integrity in Gram-negative bacteria such as *E. coli* because the drug vancomycin cannot permeate the outer membrane except if outer membrane integrity is breached by any means. Then, and only then, is sensitivity to vancomycin seen (*Hart and Silhavy, 2020*). To test whether expression of a mutant form of OmpC that cannot engage BamD in vivo would likewise diminish membrane integrity, we assayed for growth of *E. coli* strains expressing OmpC(FY) and other mutant forms of OmpC (*Figure 8F and G*). The OmpC(FY) –5 strand double mutant showed increased sensitivity to vancomycin (*Figure 8G*). The OmpC(FY) mutant does not accumulate in the outer membrane to the level found in wild-type *E. coli* (*Figure 3D*). The strain co-expressing OmpC(FY) with endogenous OmpC has diminished levels of another major porin (OmpF) as judged by BN-PAGE, but not OmpC(WT) (*Figure 8I*, see also *Figure 3*). The dimeric form of endogenous OmpF was prominently observed in both the OmpC(WT) and the OmpC(VY) double mutant cells.

## Discussion

OMPs are the major components of bacterial outer membranes, accounting for more than 50% of the mass of the outer membrane (*Horne et al., 2020*; *Jarosławski et al., 2009*). The quantitative analysis of single-cell data suggests approximately 500 molecules of major porins, such as OmpC or OmpF, are assembled into the outer membrane per minute per cell (*Benn et al., 2021*; *Lithgow et al., 2023*). This considerable task requires factors that promote the efficient and prompt assembly of OMPs in order to handle this extraordinary rate of substrate protein flux. The efficiency of assembly is necessary to fight against external cellular insult, hence the use of BamD as an OMP assembly accelerator.

In our experimental approach to assess for inhibitory peptides, specific segments of the major porin substrate OmpC were shown to interact with the BAM complex as peptidomimetic inhibitors. Results for this experimental approach went beyond expected outcomes by identifying the essential elements of the signal $\Phi xxxxxx[\Omega/\Phi]x[\Omega/\Phi]$ in β-strands other than the C-terminal strand. Discovery of conserved, internal signals in OMPs explains two previous reports where the –5 strand of BamA was

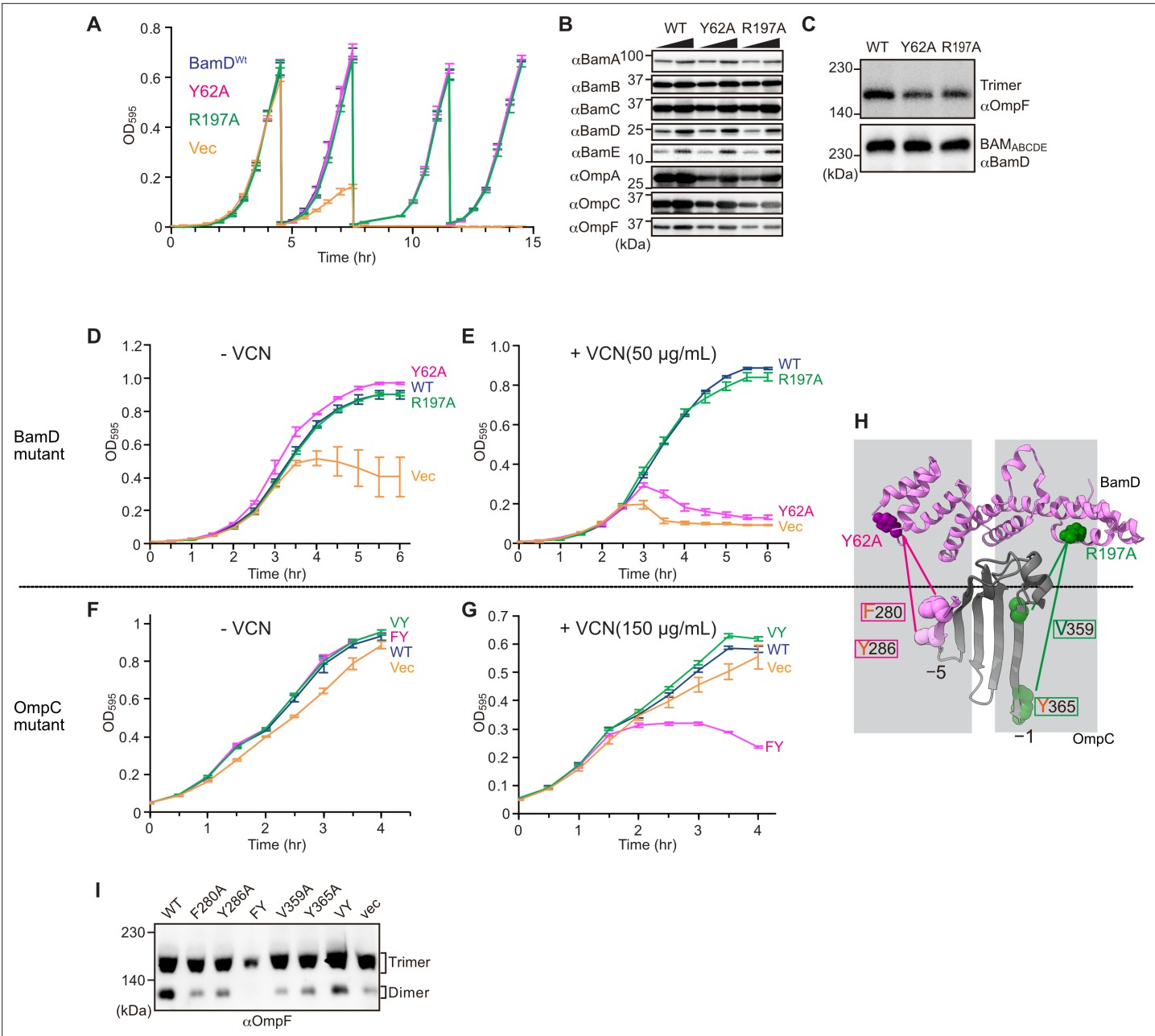

**Figure 8.** Substrate recognition by BamD is essential to maintain membrane integrity. (**A**) Strains of E. coli with a depletion of chromosomal bamD, complemented by expression of the indicated forms of BamD, were grown in restrictive media (LB supplemented with 0.5% glucose) for 4 hours, and then cultures were diluted into freash same media every 4 hours and measured growth curve. (**B**) Quantitation of steady-state protein levels of OmpF and the indicated subunits of the β-barrel assembly machinery (BAM) complex in WT, Y62A, R197A mutants were determined after SDS-PAGE and immunoblot analysis. Wedge indicates that in each case, two samples were assessed corresponding to either 4 or 12 μg of total protein. (**C**) Quantitation of the level of trimeric OmpF formed in the indicated *E. coli* microsomal membrane (EMM) assays. The immunoblot analysis is shown after Blue native (BN)-PAGE analysis and immunoblotting with antibodies recognizing OmpF (upper panel) or BamD (lower panel). (**D, E**) *E. coli* bamD depletion cells expressing mutations at residues, Y62A and R197A, in the β-signal recognition regions of BamD were grown in the presence of VCN. (**F, G**) *E. coli* cells expressing mutations to OmpC internal signal, as shown in *Figure 3*, grown in the presence of VCN. Mutations to two key residues of the internal signal were sensitive to the presence of VCN. (**H**) Structure of BamD (top) and the final five β-strands of OmpC (bottom). Gray boxes indicate relative regions of interaction between BamD and substrates. (**I**) OmpF complex assembly in vivo of cells expressing mutant OmpC. OmpF assembly was greatly reduced in cells expressing double mutations to the −5 internal β-signal.

suggested to be more similar to a β-signal motif than the C-terminal strand of BamA (*Hagan et al., 2013*; *Imai et al., 2011*) and a hidden Markov model analysis that showed that in autotransporter OMPs, a class of OMPs which includes EspP, 473/1511 of non-redundant proteins show no β-signal motif at the C-terminal strand, but instead contained a β-signal motif in the –5 strand (*Celik et al., 2012*; *Cox et al., 2010*).

The inhibitory peptides discovered here contained information that led us to define an internal signal for OMP assembly. This internal signal corresponds to the –5 strand in OmpC and is recognized by BamD. Sequence analysis shows that similar sequence signatures are present in other OMPs (*Figure 2—figure supplements 1–4*). These sequences were investigated in two further OMPs: OmpF and LamB (*Figure 2C and D*). Hidden Markov model approaches like HMMR (*Eddy, 1995*) detect conserved sequence features and tools such as MEME (*Bailey et al., 2009*) can then define a sequence motif derived from information across very many non-redundant protein sequences. The abundance of information which arises from modeling approaches and from the multitude of candidate OMPs is generally oversimplified when written as a primary structure description typical of the β-signal for bacterial OMPs (i.e. $\zeta$ xGxx[$\Omega/\Phi$]x[$\Omega/\Phi$]) (*Kutik et al., 2008*). Analysis of the few peptidomimetic peptide sequences in functional analyses, via both in vitro assays and intact *E. coli* analysis, suggests that $\Phi$xxxxx[$\Omega/\Phi$]x[$\Omega/\Phi$] is a descriptor for the internal signal, which is complimentary in function to the C-terminal β-signal that engages the BAM complex to assist OMP assembly into the outer membrane.

The β-signal motif ($\zeta$ xGxx[$\Omega/\Phi$]x[$\Omega/\Phi$]) is an eight-residue consensus, and internal signal motif is composed of a nine-residue consensus. Recent structures have shown the lateral gate of BamA interacts with a seven-residue span of substrate OMPs. Interestingly, inhibitory compounds, such as darobactin, mimic only three residues of the C-terminal side of β-signal motif. Cross-linking presented here in our study showed that BamD residues R49 and G65 cross-linked to the positions 0 and 6 of the internal signal in OmpC (*Figure 6D*). Both signals are larger than the assembly machinery's signal binding pocket, implying that the signal might sit beyond the bounds of the signal binding pocket in BamD and the lateral gate in BamA. These finding are consistent with similar observations in other signal sequence recognition events, such as the mitochondrial targeting presequence signal that is longer than the receptor groove formed by the Tom20, the subunit of the translocator of outer membrane complex (*Yamamoto et al., 2011*). The presequence has been shown to bind to Tom20 in several different conformations within the receptor groove (*Nyirenda et al., 2013*).

The current dogma in our field states that the information contained in the β-signal is important for engagement into the lateral gate of BamA (*Bitto and McKay, 2003*; *de Cock et al., 1997*; *Doyle et al., 2022*; *Doyle and Bernstein, 2019*; *Hagan et al., 2015*; *Jansen et al., 2000*; *Robert et al., 2006*; *Tomasek et al., 2020*; *Xiao et al., 2021*). This interaction initiates the insertion of a nascent OMP polypeptide into the outer membrane. By this model, BamD is considered to function by organizing the BamC and BamE subunits of the BAM complex against the core subunit BamA (*Kim et al., 2011*; *Kim et al., 2007*; *Malinverni et al., 2006*). If that is the sole role of BamD, then BamD is essential because its importance in structural organization of the BAM complex (*Hart and Silhavy, 2020*), not because of a substrate recognition function. Previous studies have shown that BamD is capable of stimulating the folding and insertion of OMPs into liposomes in the absence of BamA (*Hagan et al., 2013*; *Hagan et al., 2010*). The question remains, how is BamD capable of this function? In our present study, we show that incubation of OmpC with BamD promotes the formation of antiparallel β-strands by the formation of intramolecular disulfide cross-linking of neighboring β-strands at the C-terminus (*Figure 7B*). In EspP position 1214, located within the internal signal, was seen to cross-link to BamD (*Ieva et al., 2011*), and our intermolecular cysteine cross-linking showed that the N-terminal helices of BamD's TPR structure interact with the internal signal and the C-terminal helices interact with the β-signal (*Figure 6B and C*). Failure in this interaction was seen to stall substrate assembly within the soluble domain of the BAM complex (*Figure 4C* and *Figure 5D*). This suggests that BamD acts in an important role in an early recognition step in which both the substrate's C-terminal β-signal and internal signal engage BamD in order to be arranged into a form ready for transfer to the lateral gate of BamA (*Shen et al., 2023*; *Tomasek et al., 2020*; *Figure 9A*). This adds an additional role of the C-terminus of BamD beyond a complex stability role (*Ricci et al., 2012*; *Storek et al., 2024*). The recognition and initiation of β-strand formation by BamD is necessary to support the efficient assembly of the 500 molecules of OMPs per minute

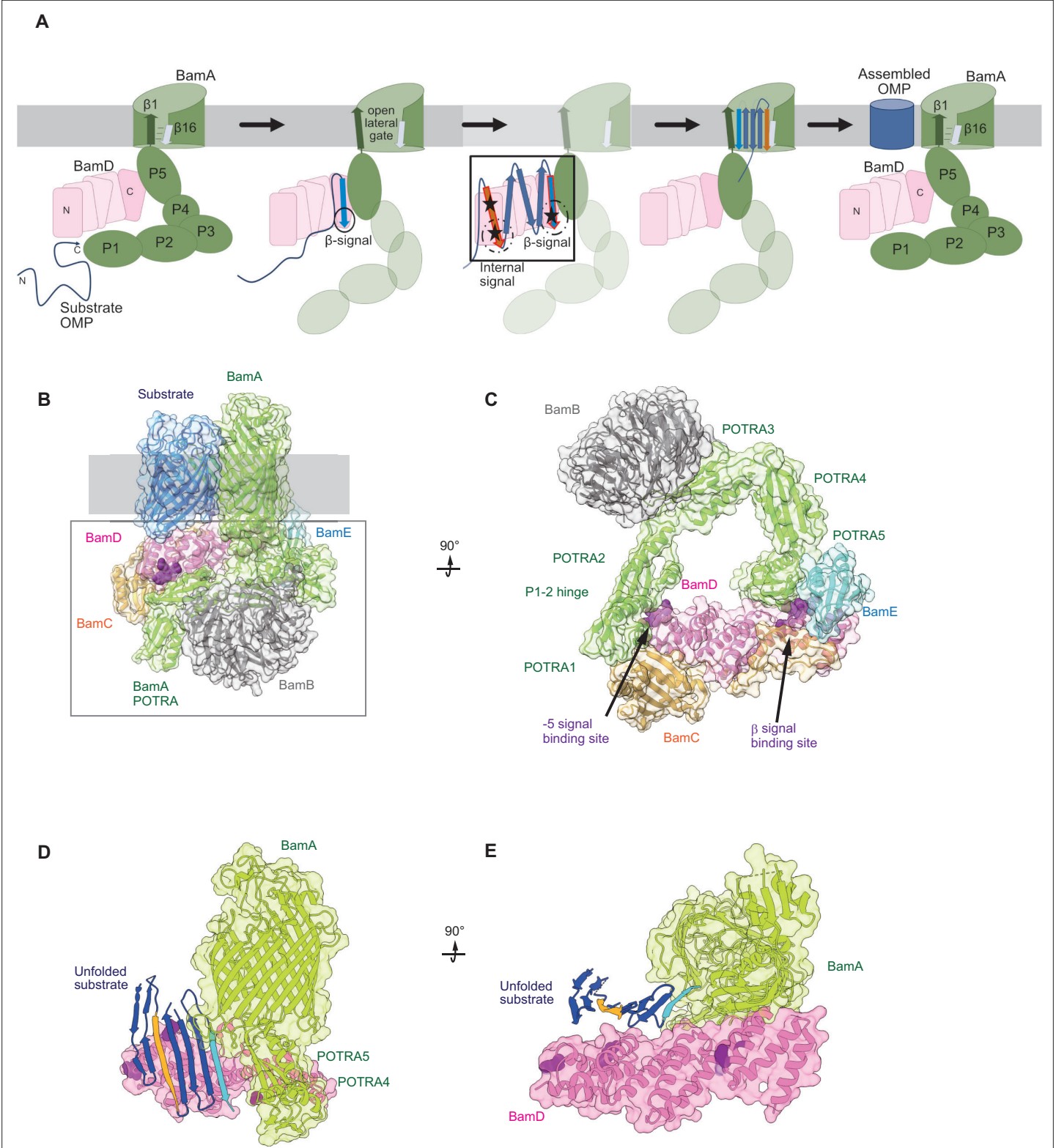

**Figure 9.** Model for the recognition of internal and canonical β-signals by the β-barrel assembly machinery (BAM) complex. (**A**) BAM complex-assisted outer membrane protein (OMP) assembly is initiated when substrate OMPs encounter the periplasmic domain of the BAM complex. The canonical β-signal (blue arrow) of the OMP engages with the C-terminal TPR motifs of BamD (pink), after which the internal β-signal (orange arrow) is marshalled by the N-terminal TPR motif. The internal signal interacts with BamD R49 and N60 (black stars) while the canonical β-signal interacts with BamD D204 (black star). The proximity of BamD substrate to POTRA5 and the organization of the C-terminal β-strands of OMPs can then stimulate the entrance to

*Figure 9 continued on next page*

*Figure 9 continued*

the lateral gate of BamA. Finally, barrel formation and release into the membrane follows as previously published. (**B**) Structure of substrate engaged BAM complex (PDBID: 6V05). (**C**) Bottom-up view of periplasmic domain of complex as shown in (**B**). Arrows indicate the position of the residues on BamD that function in binding sites for –5 signal and the β-signal. (**D**) A cavity formed by POTRA1-2 and POTRA5 of BamA and BamD is sufficient to accommodate the C-terminal five β-strands of substrate OMP. Substrates –1 strand and –5 β-strand are indicated by light blue strand and orange strand, respectively. The signal-binding region of BamD is emphasized by deep purple (PDBID; Bam complex: 7TT5 and 7TT2). (**E**) Bottom-up view of complex as shown in (**D**).

required by the outer membrane, ensuring the survival against foreign insult and supports bacterial survival (*Lithgow et al., 2023*).

Recent advances in our understanding of BAM complex function have come from structural determination of EspP engaged in assembly by the BAM complex (*Doyle et al., 2022*; *Shen et al., 2023*). The assembly intermediate of EspP engaged with the BAM complex shows the –4 strand, interacts with the R49 residue of BamD, in the N-terminal substrate-binding region, even before the –5 strand had entered into this cavity (*Doyle et al., 2022*). Spatially, this indicates that the BamD can serve to organize two distinct parts of the nascent OMP substrate at the periplasmic face of the BAM complex, either prior to or in concert with, engagement to the lateral gate of BamA. Assessing this structurally showed that the N-terminal region of BamD (interacting with the POTRA1-2 region of BamA) and the C-terminal region of BamD (interacting with POTRA5 proximal to the lateral gate of BamA) (*Bakelar et al., 2016*; *Gu et al., 2016*; *Tomasek et al., 2020*) has the N-terminal region of BamD changing conformation depending on the folding states of the last four β-strands of the substrate OMP, EspP (*Doyle et al., 2022*). The overall effect of this being a change in the dimensions of this cavity change, a change which is dependent on the folded state of the substrate engaged in it (*Figure 9B–E*). We showed that purified BamD can facilitate partial folding of an OMP and we propose that BamD captures the –1 strand and draws the –5 strand by a conformational change, thereby catalyzing the formation of hydrogen bonds to form β-hairpins in neighboring strands, to prime the substrate for insertion into the outer membrane by the action of BamA. After OMP assembly, all elements of the internal signal are positioned such that they face into the lipid phase of the membrane. This observation may be a coincidence, or may be utilized by the BAM complex to register and orientate the lipid facing amino acids in the assembling OMP away from the formative lumen of the OMP. Amino acids at position 6, such as Y286 in OmpC, are not only component of the internal signal for binding by the BAM complex, but also act in structural capacity to register the aromatic girdle for optimal stability of the OMP in the membrane.

# Materials and methods

**Key resources table**

| Reagent type (species) or resource | Designation | Source or reference | Identifiers | Additional information |
|---|---|---|---|---|
| Gene (*Escherichia coli*) | Mg1655 | Lithgow lab | | Template for cloning |
| Strain, strain background (*Escherichia coli*) | BL21(DE3)* | Invitrogen | | Protein expression |
| Strain, strain background (*Escherichia coli*) | bamD depletion | This paper | bamD depletion | For analysis of bamD function |
| Recombinant DNA reagent | pTnT-f (primer) | *Thewasano et al., 2023* | PCR primers for in vitro transcription/translation | ACTTAATA CGACTCAC TATAGGCT A |
| Recombinant DNA reagent | pTnT-r (primer) | *Thewasano et al., 2023* | PCR primer for in vitro transcription/translation | GGATCCAA AAAACCCC TCAAGACC C |
| Recombinant DNA reagent | pET22-BamAL4H8 (Plasmid) | *Ding et al., 2020* | | For protein expression for NR |

*Continued on next page*

*Continued*

| Reagent type (species) or resource | Designation | Source or reference | Identifiers | Additional information |
|---|---|---|---|---|
| Recombinant DNA reagent | pHIS-BamD (Plasmid) | *Chen et al., 2021* | | For protein expression for NR |
| Recombinant DNA reagent | pET15b-OmpC (Plasmid) | This paper | | For analysis of in vitro folding of OmpC |
| Recombinant DNA reagent | pBAD-FLOmpC (Plasmid) | This paper | | For analysis of OmpC assembly in vivo |
| Recombinant DNA reagent | pHIS-BamD (X) amber (Plasmid) | This paper | | For in vitro photo-cross-linking |
| Recombinant DNA reagent | pHIS-BamD (X) cys (Plasmid) | This paper | | For in vitro disulfide cross-linking |
| Recombinant DNA reagent | pAp-BamD (Plasmid) | This paper | | For analysis of bamD function |
| Antibody | Anti-DYKDDDDK (Mouse monoclonal) | FUJIFILM-Wako | Cat# 014-22383 | SDS-WB(1:5000) BN-WB(1:5000) |
| Antibody | Anti-BamA (Rabbit polyclonal) | Lithgow lab | | SDS-WB(1:10,000) BN-WB(1:10,000) |
| Antibody | Anti-BamB (Rabbit polyclonal) | Lithgow lab | | SDS-WB(1:10,000) BN-WB(1:5000) |
| Antibody | Anti-BamC (Rabbit polyclonal) | Shiota lab | | SDS-WB(1:20,000) Ab-Shift(1:100) |
| Antibody | Anti-BamD (Rabbit polyclonal) | Lithgow lab | | SDS-WB(1:10,000) BN-WB(1:5000) |
| Antibody | Anti-BamE (Rabbit polyclonal) | Lithgow lab | | SDS-WB(1:10,000) |
| Antibody | Anti-OmpF (Rabbit polyclonal) | Lithgow lab | | SDS-WB(1:10,000) BN-WB(1:5000) |
| Antibody | Anti-OmpA (Rabbit polyclonal) | Shiota lab | | SDS-WB(1:10,000) |
| Antibody | Anti-OmpC (Rabbit polyclonal) | Lithgow lab | | SDS-WB(1:10,000) |
| Antibody | Anti-Mouse IgG-Peroxidase (Produced in goat) | Sigma-Aldrich | Cat# A4416 | 1:20,000 |
| Antibody | Anti-Rabbit IgG-Peroxidase (Produced in goat) | Sigma-Aldrich | Cat# A6154 | 1:20,000 |
| Peptide, recombinant protein | Peptides containing a partial sequence of OmpC | Mimotopes | Custom peptide 'Pepset' | |
| Commercial assay or kit | Rabbit Reticulocyte Lysate, Nuclease-Treated | Promega | Cat# L4960 | |
| Commercial assay or kit | EXPRESS35S Protein Labeling Mix | Perkin Elmer | Cat# NEG072 | |
| Commercial assay or kit | SP6 RNA Polymerase | Clontech | Cat# 2520A | |
| Commercial assay or kit | Can Get Signal Immunoreaction Enhancer Solution | TOYOBO | Cat# NKB-101 | |
| Commercial assay or kit | Chemi-Lumi One L | Nacalai tesque | Cat# #07880 | 0.03 ml/cm$^2$ |

*Continued on next page*

*Continued*

| Reagent type (species) or resource | Designation | Source or reference | Identifiers | Additional information |
|---|---|---|---|---|
| Chemical compound, drug | *n*-Dodecyl-β-D-maltoside | Dojindo | Cat# #34106161 | |
| Chemical compound, drug | *p*-Benzoyl-L-phenylalanine | Watanabe Chemical Industries | Cat# #H00146 | |
| Other | Gold-coated Si-Wafer | Yamanaka Hutech | Custom ordered | Gold coat was performed by Melbourne Centre for Nanofabrication |

*E. coli* strains and growth conditions.

*E. coli* strains used in this study are listed in **Supplementary file 1**. All strains were grown in LB (1% tryptone, 1% NaCl, and 0.5% yeast extract). *E. coli* strains used for the in vivo photo-cross-linking experiments were grown in XB (1% tryptone, 1% NaCl, and 0.1% yeast extract). Unless otherwise specified, ampicillin (amp; 50 mg/ml), chloramphenicol (Cm; 10 mg/ml), or kanamycin (Kan, 30 mg/ml) were added to media for plasmid selection.

To create the BamD depletion strain, we amplified four DNA fragments encoding 540 to 40 base pairs of the 5′ UTR of BamD (primers BamDSO-1 and -2), kanamycin cassette (primers BamDSO-3 and -4), AraC-pBAD (primers BamDSO-5 and -6), and the first 500 bp of BamD (primers BamSO-7 and -8), respectively (*Figure 7—figure supplement 3*). *Supplementary file 2* describes the pair of primers and template DNA to amplify each DNA fragment. These four DNA fragments were combined by the overlap PCR method and transformed into MC4100a harboring pKD46 encoding $\lambda$ Red recombinase (*Datsenko and Wanner, 2000*). Transformants were selected in the presence of kanamycin and 0.1% (wt/vol) L-arabinose.

To culture BamD depletion strains a single colony from each BamD depletion strains harboring pBamD variants was inoculated into LB (+ amp + Kan + 0.02% [wt/vol] L-arabinose) and cultured for 10 hr at 37°C. The saturation culture was shifted into the LB (+ amp + Kan) and incubated for another 10 hr at 37°C. The saturation culture was diluted into LB (+ amp + kan + 0.5% [wt/vol] D-glucose) to an $OD_{600}=0.02$, and incubated at 37°C until cell density reached $OD_{600}=0.8$. A second dilution in the same media to an $OD_{600}=0.02$ was performed, and incubated for 2.5 hr at 37°C.

## Plasmids

Plasmids and primers used in this study are listed in *Supplementary file 3* to *Supplementary file 7*. Cloning of specific target gene ORFs into specified vectors was performed via SLiCE in vitro recombination (*Okegawa and Motohashi, 2015*).

## Protein purification

Plasmids encoding each subunit of the BAM complex and OmpC genes (including a His$_6$-tag) were transformed into BL21(DE3)* cells (Invitrogen). BamD and OmpC variant transformants were grown in LB (+ amp) at 37°C to an $OD_{600}=0.5$. Expression was induced by adding 1 mM IPTG and the cells grown at 37°C for 3 hr. For BamD containing BPA mutations, MC4100 cells were transformed with pHIS-BamD-(X)-amber and pEVOL-BpF45. Transformants were cultured in LB with antibiotics to an $OD_{600}=0.5$, then supplemented with 1 mM BPA, 0.05% (wt/vol) L-arabinose and 1 mM IPTG to induce expression. Cells were grown at 37°C for a further 3 hr.

Cells were pelleted by centrifugation at 6000 × *g* for 10 min. Pellets were resuspended in Buffer A (150 mM NaCl, 20 mM NaPO$_4$) and lysed via sonication. Lysates were clarified by low-speed centrifugation of 1100 × *g* for 10 min at 4°C. For purification of BamD variant proteins, supernatants were again clarified by high-speed centrifugation, 17,000 × *g* for 15 min at 4°C. High-speed clarified supernatants were then applied to Ni$^{2+}$-NTA-affinity column chromatography. Ni$^{2+}$-NTA resin were washed by Buffer A with 50 mM imidazole and proteins were eluted in Buffer A containing an imidazole step gradient (100–400 mM). Peak fractions were collected as the purified fraction. To purify OmpC, the pellets containing OmpC in inclusion bodies were resuspended in Buffer A containing 0.5% Triton X-100 and centrifuged for 11,600 × *g* for 10 min at 4°C. Inclusion bodies were denatured in Buffer A with 6 M urea and mixed at 25°C for 90 min, followed by centrifugation of 11,600 × *g* for 10 min at 4°C. The filtered supernatant was applied to Ni$^{2+}$-NTA-affinity column chromatography. The bound OmpC-His$_6$ was washed with Buffer A (+6 M urea, 50 mM imidazole) and eluted with Buffer A containing 6 M urea and 200 mM imidazole. Purified proteins were then diluted with glycerol (final concentration of 10%),

snap-frozen in liquid nitrogen, and stored in –80°C. BamA was purified as detailed previously (**Ding et al., 2020**).

## Isolation of the EMM fraction

*E. coli* crude membrane fraction was isolated as previously described (**Gunasinghe et al., 2018**). In brief, pelleted cells were resuspended in sonication buffer (50 mM Tris-HCl pH 7.5, 150 mM NaCl, 5 mM EDTA) and lysed via sonication on ice. After low-speed centrifugation (1100 × $g$, 5 min, 4°C), supernatant was centrifuged again (15,000 × $g$, 10 min, 4°C). Pelleted membranes were resuspended in SEM buffer (250 mM sucrose, 10 mM MOPS-KOH pH 7.2, 1 mM EDTA) and flash-frozen. Protein concentrations of EMM were calculated by $OD_{280}$ measurements of 100 times diluted in 0.6% SDS ($OD_{280}$=0.21 was set to be 10 mg/ml).

## EMM assembly assay

The EMM assembly assay has been previously described (**Gunasinghe et al., 2018**; **Thewasano et al., 2023**). The method in brief is as follows: substrates were prepared via in vitro transcription by SP6-RNA polymerase followed by in vitro translation in rabbit reticulocyte lysate supplemented with $^{35}$S-methionine. EMMs were resuspended in Assembly Assay buffer (10 mM MOPS-KOH pH 7.2, 2.5 mM $KH_2PO_4$, 250 mM sucrose, 15 mM KCl, 5 mM $MgCl_2$, 2 mM methionine, 5 mM DTT, 1% wt/vol BSA, 0.09% vol/vol Triton X-100), and incubated with $^{35}$S-labeled substrate at 30°C for indicated time points. Assembly reactions were halted by moving to ice for 5 min. The EMMs were then harvested via centrifugation at 150,000 × $g$ for 5 min at 4°C and washed with SEM buffer. After SEM buffer wash, trimeric protein-assembled EMM pellets were solubilized in 1.5% DDM containing BN-PAGE Lysis Buffer (25 mM imidazole-HCl pH 7.0, 50 mM NaCl, 50 mM 6-aminohexanoic acid, 1 mM EDTA, 7.5% [wt/vol] glycerol) on ice for 20 min. Solubilized proteins were clarified by centrifugation of 15,000 × $g$ for 10 min at 4°C, followed by BN-PAGE analysis as described previously (**Gunasinghe et al., 2018**; **Thewasano et al., 2023**).

Peptide library was synthesized by MIMOTOPES, and resuspended in DMSO at a concentration of 30 mM. The peptide inhibition screen of EspP was performed as stated above, but with the addition of 0.1 mM inhibitory peptide to the Assembly Assay buffer and incubated with EMMs for 5 min prior to the addition of $^{35}$S-labeled substrate protein. After harvesting, EMMs were treated with 100 μg/ml Proteinase K on ice for 20 min. Digestion was halted by adding 2 mM PMSF. The EMMs were then collected via centrifugation at 150,000 × $g$ for 5 min at 4°C and washed with SEM buffer. EspP assembled with EMM proteins were analyzed by SDS-PAGE and radio-imaging. EMM assembly of trimeric proteins was tested with the addition of 0.25 mM peptide and analyzed as above.

## Sequence conservation analysis

For conservation analysis, we obtained homologous sequences of representative structure-known β-barrel OMPs from reference bacterial proteomes in UniProt (The UniProt Consortium, 2019) using phmmer and jackhammer HMMER 3.2.1 (http://hmmer.org). The best hit sequences in organisms which also contained a BamD homolog were used. We identified 31, 94, 25, 19, 27, 44, 43, 23, 42, 22, 132, 46, 58, 68, 39, 245, 261,115, 187, 250, and 74 sequences for OmpA, OmpW, PagP, OmpX, OmpT, EspP, Hbp, NanC, OMPLA, Tsx, FadL, OmpC, OmpF, PhoE, LamB, BtuB, Cir, FecA, FepA, FhuA and PapC, respectively (sequence identity less than 60%, expect for OmpX. In case of OmpX, sequence identity is less than 80% due to the small number of sequences). Multiple alignments were generated by MAFFT (**Katoh et al., 2019**) and then sequence logos of –5th transmembrane strands were created using WebLogo 3 (**Crooks et al., 2004**).

## In vivo assembly analysis of FLAG-OmpC

We introduced a FLAG epitope tag into the N-terminal region of OmpC, immediately behind the cleavage site for the SEC-signal sequence (**Rapoport, 2007**), to distinguish between endogenous OmpC and the OmpC variants in the OmpC deletion strain background which has issues of diminished OM integrity. The expression of FLAG-OmpC mutants was controlled by using a plasmid-based pBADara promoter cassette, to avoid the chronic effect of constitutively expressed mutant proteins (**Figure 3A**). We transformed pBAD-FLOmpC variants into MC4100A. Transformants were cultured in

LB (+ amp) until OD$_{600}$=0.5 and then protein expression induced by adding 0.1% arabinose for 3 hr. *E. coli* cells were harvested and total cell lysate or EMMs prepared accordingly.

## Antibody shift assay

The variation of BN-PAGE analysis referred to as an antibody shift assay was performed as previously described with modification (*Shiota et al., 2012*). In brief, solubilized EMMs were incubated with 3 µl of anti-BamC antibodies for 45 min at 4°C. Samples were clarified via centrifugation at 13,000 × *g* for 10 min at 4°C, and subjected to BN-PAGE.

## Urea extraction of intermediate

Once the Assembly Assay was performed, to obtain intermediate complexes, lysates were washed with SEM, the EMMs resuspended in SEM containing 6 M urea and incubated at 37°C for 1 hr, after which 50 mM Tris HCl pH 8.0 was added. Membrane fractions were collected via high-speed centrifugation (100,000 × *g*, 45 min, 25°C) and washed twice with SEM buffer. The membrane was solubilized for 20 min at 4°C in 1.5% DDM containing BN-PAGE lysis buffer and subjected to BN-PAGE analysis.

## Neutron reflectometry

Neutron reflection was carried out on the SOFIA reflectometer at the Materials and Life Science Experimental Facility (MLF) at J-PARC, Japan (*Mitamura et al., 2013*; *Yamada et al., 2011*). The reflectivity at the interface between the Si substrate and water solution was measured at three incident angles (0.3°, 0.75°, and 1.8°). The neutrons were introduced from the substrate side to illuminate the interface and the reflection intensity was normalized by the transmission intensity through the substrate to achieve the reflectivity with the Q range of up to 0.2 Å$^{-1}$. Neutron reflectivity, denoted as R, is the ratio of the incoming to the exiting neutron beams. It's calculated based on the momentum transfer Q, which is defined by the formula Q=4π sinθ/$\lambda$, where θ represents the angle of incidence and $\lambda$ stands for the neutron wavelength. The approximate value of R(Q) can be expressed as R(Q)=16π$^2$/Q$^2$|$\rho$(Q)|2, where R(Q) is the one-dimensional Fourier transform of $\rho$(z), which is the SLD distribution perpendicular to the surface. SLD is calculated by dividing the sum of the coherent neutron scattering lengths of all atoms in a sample layer by the volume of that layer. Consequently, factors such as thickness, volume fraction, and interface roughness of the samples significantly influence the intensity of the reflected beams.

Analysis of the NR profiles was performed using the MOTOFIT analysis software (*Nelson, 2010*). The software uses a conventional method, optical matrix method (*Nelson, 2006*), to model the reflectivity profiles. Briefly, the layer is divided into several sublayers and then a characteristic matrix is evaluated for each sublayer, from which the whole reflectivity can be calculated exactly. The same layer with multiple datasets with three different isotopic contrasts is fitted simultaneously until the satisfactory fit is obtained (smallest $\chi^2$ value in the genetic algorithm). All models include a 15 Å oxide layer on the surface of the silicon substrates and subsequently there are three layers of Ni, gold, and Ni-NTA on the top of oxide layer. Protein layers will assemble outward from the Ni-NTA layer.

The best-fit model from MOTOFIT gives information about the thickness, density, and roughness of each layer. When a layer is composed of just two components, a chemical species or its fragment s and water w, the SLD is given by:

$$\rho_{layer} = \varphi\rho_s + (1 - \varphi)\rho_w$$

where $\rho_w$ and $\rho_s$ are the SLDs of the two components, respectively, and φ is the volume fraction of species s in the layer. If more than one chemical species is present, then $\rho_s$ is determined by the volume fraction of each species in the layer.

## OmpC in vitro folding

Urea-denatured OmpC and OmpC variants were purified as stated above. Protein was resuspended in Refolding Buffer (10 mM phosphate buffer pH 8.0, 2 mM EDTA, 1 mM glycine) to a final concentration of 5 µM. DDM was added to a final concentration of 3.4 mM. Samples were incubated at 37°C for 0, 1, 3, 6, 8, 12, 24, and 36 hr. SDS-loading dye was added and each sample was split into two tubes (one to be boiled, one at RT for 10 min). OmpC folding was analyzed via SDS-PAGE.

## Ni-NTA substrate pull-down

Purified BamA or BamD was resuspended in Buffer A to a final concentration of 10 μM. Specific peptides or DMSO alone was added (final concentration 0.35 mM) and incubated rocking at 4°C for 1 hr. To each tube, 0.5 μl of $^{35}$S-OmpC was added and vortexed vigorously, 10% of the total volume was removed to assay as 'input' for each sample. Samples were then incubated at 4°C with rocking for 30 min. Ni-NTA agarose beads were added to each sample, incubated at 4°C with rocking for an additional 30 min then washed with Buffer A containing 20 mM imidazole. BAM proteins were eluted from the Ni-NTA resin using Buffer A with 400 mM imidazole. Proteins were TCA precipitated and subjected to analysis by SDS-PAGE and imaged via radio-imaging.

## Disulfide cross-linking

BamD-OmpC cysteine cross-linking was performed using 50 μM purified single-cysteine BamD mutant variants incubated with 8 μl of single-cysteine OmpC variants translated by the rabbit reticulocyte lysate in 30 μl of Buffer A at 25°C for 2 min. An oxidizing reaction mixture of 1 μM CuSO$_4$, 4 mM methionine, and 4 mM cysteine was added and incubated at 25°C for 5 min. Proteins were precipitated by TCA precipitation. Proteins were washed with 100% acetone and solubilized in 1% SDS buffer. Solubilized proteins were diluted by 0.5% Triton X-100 buffer and purified with Ni-NTA via the tagged BamD-H6. BamD and disulfide cross-link products were eluted with 400 mM imidazole containing 0.5% Triton X-100 buffer. Eluted proteins were concentrated by TCA precipitation and solubilized with SDS-PAGE loading dye with or without β-Me.

The OmpC intrachain disulfide bond assay was performed using 1 μl $^{35}$S-OmpC double cysteine variants translated in the presence of 1 mM DTT. OmpC variants were diluted into 100 μl Buffer A then incubated with 50 μM BamD at 32°C for 30 min. 10 μM CuSO$_4$ was added to samples and incubated for an additional 2 min, followed by the addition of SDS-PAGE loading dye with or without 5 mM DTT and 1% β-Me. Proteins were resolved via SDS-PAGE and imaged as stated above.

## In vivo and in vitro photo-cross-linking

For in vivo photo-cross-linking, MC4100 strain was transformed with pEVOL-pBpF, pTnT-BamAp-BamD-His8, and pET-FLAG-OmpC, with respective antibiotics. A single colony was used to inoculate a culture in photo-cross-linking media (1% NaCl, 1% tryptone, 0.1% extract yeast dried, 1 mM BPA) for 3 hr at 37°C in the dark. Afterward, arabinose (0.1% final concentration) was added and further incubated for 2.5 hr. Cultures were divided with one half UV-irradiated for 7 min at RT and the other half not. Cells were pelleted and solubilized with 1% SDS buffer (50 mM Tris-HCl pH 8.0, 150 mM NaCl, 1% SDS). His-tagged BamD containing BPA and its cross-linked products were purified by Ni-NTA agarose. Cross-linked products were identified via immunoblotting with anti-BamD and anti-FLAG antibodies.

In vitro photo-cross-linking was performed with 32 μg of purified BamD-His$_8$ variants containing BPA diluted in Buffer A. 40 μl of $^{35}$S-OmpC was added to each tube and incubated at 25°C for 10 min in the dark. Each mixture was transferred to the lid of a microtube and irradiated with UV using B-100AP (UVP) for 7 min on ice. 10% (wt/vol) SDS solution was added to a final concentration of 1%, then incubated at 95°C for 10 min to denature proteins. Protein mixtures were diluted with 0.5% Triton X-100 buffer (50 mM Tris-HCl pH 7.0, 150 mM NaCl, 0.5% Triton X-100), and purified on Ni-NTA agarose beads. Elute fractions were concentrated by TCA precipitation and analyzed by SDS-PAGE and radio-imaging.

## Statistical analysis

For all densitometry measurements, we utilized 'imagequant' quant (ImageQuant TL, Cytiva) and statistical analysis and graphing was performed with Microsoft Excel software.

## Acknowledgements

We thank the members of the Shiota, Shen, and Lithgow labs for discussion and critical comments on the manuscript. This work was supported by JSPS KAKENHI to TS (19K16077, 18KK0197, 18H06052, 22K12672, and 21KK0126), to EMG (21K15043), and to KI (18K11543 and 21H03551), JST FOREST Program to TS (JPMJFR2064), AMED Research Support Project for Life Science and

Drug Discovery (Basis for Supporting Innovative Drug Discovery and Life Science Research [BINDS]) (23ama121029j0002) to KI, Australian Research Council (DP160100227) to TL, and National Health and Medical Research Council (CDF1106798) to HHS. The following grants are also acknowledged; a grant from the Ichiro Kanehara Foundation for the Promotion of Medical Sciences and Medical Care, Waksman Foundation of Japan, Tokyo Biochemical Research Foundation, Sumitomo Foundation, Naito Foundation, Uehara Memorial Foundation, The Shinnihon Foundation of Advanced Medical Treatment Research, and Noguchi Institute (to TS) and National Health and Medical Research Council (2016330) to TL. The NR experiment at the J-PARC MLF was performed under a user program (Proposal No. 2019B0364). We thank NL Yamada for support of NR measurements, and H Nishitoh for access to instruments (for TS). Radiation experiments were supported by the RI Kiyotake of the Frontier Science Research Center, University of Miyazaki.

## Additional information

### Funding

| Funder | Grant reference number | Author |
|---|---|---|
| Japan Society for the Promotion of Science | 21K15043 | Edward M Germany |
| Japan Agency for Medical Research and Development | 23ama121029j0002 | Kenichiro Imai |
| Japan Society for the Promotion of Science | 18K11543 | Kenichiro Imai |
| Japan Society for the Promotion of Science | 21H03551 | Kenichiro Imai |
| National Health and Medical Research Council | CDF1106798 | Hsinhui Shen |
| Australian Research Council | DP160100227 | Trevor Lithgow |
| Japan Science and Technology Agency | 10.52926/jpmjfr2064 | Takuya Shiota |
| Japan Society for the Promotion of Science | 19K16077 | Takuya Shiota |
| Japan Society for the Promotion of Science | 18KK0197 | Takuya Shiota |
| Japan Society for the Promotion of Science | 18H06052 | Takuya Shiota |
| Japan Society for the Promotion of Science | 22K12672 | Takuya Shiota |
| Japan Society for the Promotion of Science | 21KK0126 | Takuya Shiota |
| National Health and Medical Research Council | 2016330 | Trevor Lithgow |
| Ichiro Kanehara Foundation for the Promotion of Medical Sciences and Medical Care | | Takuya Shiota |
| Waksman Foundation of Japan | | Takuya Shiota |
| Tokyo Biochemical Research Foundation | | Takuya Shiota |
| Sumitomo Foundation | | Takuya Shiota |

| Funder | Grant reference number | Author |
|---|---|---|
| Naito Foundation | | Takuya Shiota |
| Uehara Memorial Foundation | | Takuya Shiota |
| Shinnihon Foundation of Advanced Medical Treatment Research | | Takuya Shiota |

The funders had no role in study design, data collection and interpretation, or the decision to submit the work for publication.

## Author contributions

Edward M Germany, Conceptualization, Resources, Data curation, Formal analysis, Funding acquisition, Validation, Investigation, Methodology, Writing – original draft, Writing – review and editing; Nakajohn Thewasano, Data curation, Formal analysis, Validation, Investigation, Methodology; Kenichiro Imai, Data curation, Funding acquisition, Investigation, Methodology, Writing – review and editing; Yuki Maruno, Yue Ding, Data curation, Formal analysis, Validation; Rebecca S Bamert, Christopher J Stubenrauch, Writing – original draft, Writing – review and editing; Rhys A Dunstan, Kher Shing Tan, Resources; Yukari Nakajima, Resources, Data curation, Validation; XiangFeng Lai, Formal analysis, Validation, Writing – review and editing; Chaille T Webb, Writing – original draft; Kentaro Hidaka, Resources, Data curation; Hsinhui Shen, Data curation, Formal analysis, Supervision, Funding acquisition, Validation, Investigation, Methodology, Writing – original draft, Project administration, Writing – review and editing; Trevor Lithgow, Supervision, Funding acquisition, Investigation, Writing – original draft, Writing – review and editing; Takuya Shiota, Conceptualization, Resources, Data curation, Formal analysis, Supervision, Funding acquisition, Validation, Investigation, Visualization, Methodology, Writing – original draft, Project administration, Writing – review and editing

## Author ORCIDs

Edward M Germany ⓘ http://orcid.org/0000-0003-3981-0185
Yuki Maruno ⓘ http://orcid.org/0009-0001-4905-0602
Christopher J Stubenrauch ⓘ http://orcid.org/0000-0003-4388-3184
Hsinhui Shen ⓘ http://orcid.org/0000-0002-8541-4370
Takuya Shiota ⓘ http://orcid.org/0000-0003-3004-4541

Joint Public Review: https://doi.org/10.7554/eLife.90274.3.sa1
Author Response https://doi.org/10.7554/eLife.90274.3.sa2

# Additional files

## Supplementary files

- Supplementary file 1. Bacterial strains.
- Supplementary file 2. Primers.
- Supplementary file 3. Primers for *p*-benzoyl-L-phenylalanine (BPA).
- Supplementary file 4. Plasmids.
- Supplementary file 5. Plasmids for recombinants.
- Supplementary file 6. Plasmids in vivo.
- Supplementary file 7. Structural data.
- Supplementary file 8. Characterization of BamA.
- Supplementary file 9. Characterization of BamAD.
- Supplementary file 10. Characterization of BamAD-OmpC.
- MDAR checklist

## Data availability

All data are available in the main text or the supplementary materials (Figure 5—source data 1–3).

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
