## [Editor Report · eLife assessment]

This **important** study reports the identification of a new amino acid sequence motif (i.e., "internal beta-signal") on outer membrane proteins, which is recognized by beta-assembly machinery in gram-negative bacteria. The authors carried out rigorous experiments, providing **compelling** evidence in support of their conclusions. This work significantly advances our understanding of the biogenesis of outer membrane proteins.

---

## [Referee Report · Joint Public Review]

The biogenesis of outer membrane proteins (OMPs) into the outer membranes of Gram-negative bacteria is not fully understood, particularly client recognition and insertion by the conserved beta-assembly machinery (BAM), which is itself integrated in the outer membranes. So far, the last strand of an OMP, referred to as the beta-signal, has been known as a primary recognition motif by BAM. Here, authors have identified additional sequence motifs that are located in the upstream of the last strand.

Here, authors carried out rigorous biochemical, biophysical, and genetic approaches to prove that the newly identified internal motifs are critical to the assembly of outer membrane proteins as well as to the interaction with the BAM complex. The identification of important regions on the substrates and Bam proteins during biogenesis is an important contribution that gives clues to the path substrates take en route to the membrane. Assessing the effect of the internal motifs in the assembly of model OMPs in the absence (in vitro) and presence (in vitro and in vivo) of BAM machinery aids a precise definition of the role of the motifs, solidifying the conclusions.

The initial reviews raised several concerns:

1. Strengthening the claim that the recognition of the internal signal by BAM is mediated by BamA and BamD via specific interactions.

2. Justification of the rationale of the peptide inhibition assays as a primary tool to identify significant recognition motifs.

3. More careful interpretation of the mutational effects on OMP assembly - that is, discerning the impairment of BAM-nascent polypeptide chain interaction from the impairment of intrinsic folding.

4. Providing further clarification of the discrepancy between in vitro assay and in vivo assay regarding the effect of the mutation Y286A on OMP assembly.

5. More elaboration on the rationale, interpretation, and representation of neutron refractory data.

6. An explanation is lacking why the strain with BamD R197A does not display VCN sensitivity in contrast to the strain with BamD Y62A.

Those concerns were well addressed in the revised manuscript in a rigorous manner.

Overall, this study comprehensively addresses an important question in the field. The notion that additional signals assist in biogenesis is a novel concept that has been tested and verified at least for a subset of model OMPs in this study. The generalization of the conclusion awaits a further proof of the concept.

---

## [Author Response]

The following is the authors’ response to the original reviews.

**Reviewer #1 (Public Review):**
The biogenesis of outer membrane proteins (OMPs) into the outer membranes of Gram-negative bacteria is still not fully understood, particularly substrate recognition and insertion by beta-assembly machinery (BAM). In the studies, the authors present their studies that in addition to recognition by the last strand of an OMP, sometimes referred to as the beta-signal, an additional signal upstream of the last strand is also important for OMP biogenesis.Strengths:1. Overall the manuscript is well organized and written, and addresses an important question in the field. The idea that BAM recognizes multiple signals on OMPs has been presented previously, however, it was not fully tested.1. The authors here re-address this idea and propose that it is a more general mechanism used by BAM for OMP biogenesis.1. The notion that additional signals assist in biogenesis is an important concept that indeed needs fully tested in OMP biogenesis.1. A significant study was performed with extensive experiments reported in an attempt to address this important question in the field.1. The identification of important crosslinks and regions of substrates and Bam proteins that interact during biogenesis is an important contribution that gives clues to the path substrates take en route to the membrane.Weaknesses:Major critiques (in no particular order):1. The title indicates 'simultaneous recognition', however no experiments were presented that test the order of interactions during OMP biogenesis.

We have replaced the word “Simultaneous” with “Dual” so as not to reflect on the timing of the recognition events for the distinct C-terminal signal and -5 signal.

1. Aspects of the study focus on the peptides that appear to inhibit OmpC assembly, but should also include an analysis of the peptides that do not to determine this the motif(s) present still or not.

We thank the reviewer for this comment. Our study focuses on the peptides which exhibited an inhibitory effect in order to elucidate further interactions between the BAM complex and substrate proteins, especially in early stage of the assembly process. In the case of peptide 9, which contains all of our proposed elements but did not have an inhibitory effect, there is the presence of an arginine residue at the polar residue next to hydrophobic residue in position 0 (0 Φ). As seen in Fig S5, S6, and S7, there are no positively charged amino acids in the polar residue positions in the -5 or last strands. This might be the reason why peptide 9, as well as peptide 24, the β-signal derived from the mitochondrial OMP Tom40 and contains a lysine at the polar position, did not display an inhibitory effect. Incorporating the reviewer's suggestions might elucidate conditions that should not be added to the elements, but this is not the focus of this paper and was not discussed to avoid complicating the paper.

1. The β-signal is known to form a β-strand, therefore it is unclear why the authors did not choose to chop OmpC up according to its strands, rather than by a fixed peptide size. What was the rationale for how the peptide lengths were chosen since many of them partially overlap known strands, and only partially (2 residues) overlap each other? It may not be too surprising that most of the inhibitory peptides consist of full strands (#4, 10, 21, 23).

A simple scan of known β-strands would have been an alternative approach, however this comes with the bias of limiting the experiments to predicted substrate (strand) sequences, and it presupposes that the secondary structure element would be formed by this tightly truncated peptide.

Instead, we allowed for the possibility that OMPs meet the BAM complex in an unfolded or partially folded state, and that the secondary structure (β-strand) might only form via β-argumentation after the substrate is placed in the context of the lateral gate. We therefore used peptides that mapped right across the entirety of OmpC, with a two amino acid overlap.

To clarify this important point regarding the unbiased nature of our screen, we have revised the text:

(Lines 147-151) "We used peptides that mapped the entirety of OmpC, with a two amino acid overlap. This we considered preferable to peptides that were restricted by structural features, such as β-strands, in consideration that β-strand formation may or may not have occurred in early-stage interactions at the BAM complex."

1. It would be good to have an idea of the propensity of the chosen peptides to form β-stands and participate in β-augmentation. We know from previous studies with darobactin and other peptides that they can inhibit OMP assembly by competing with substrates.

We appreciate the reviewer's suggestion. However, we have not conducted biophysical characterizations of the peptides to calculate the propensity of each peptide to form β-stands and participate in β-augmentation. The sort of detailed biophysical analysis done for Darobactin (by the Maier and Hiller groups, The antibiotic darobactin mimics a β-strand to inhibit outer membrane insertase Nature 593:125-129) was a Nature publication based on this single peptide. A further biophysical analysis of all of the peptides presented here goes well beyond the scope of our study.

1. The recognition motifs that the authors present span up to 9 residues which would suggest a relatively large binding surface, however, the structures of these regions are not large enough to accommodate these large peptides.

The β-signal motif (ζxGxx[Ω/Φ]x[Ω/Φ]) is an 8-residue consensus, some of the inhibitory peptides include additional residues before and after the defined motif of 8 residues, and the lateral gate of BamA has been shown interact with a 7-residue span (eg. Doyle et al, 2022). Cross-linking presented in our study showed BamD residues R49 and G65 cross-linked to the positions 0 and 6 of the internal signal in OmpC (Fig. 6D).

We appreciate this point of clarification and have modified the text to acknowledge that in the final registering of the peptide with its binding protein, some parts of the peptide might sit beyond the bounds of the BamD receptor’s binding pocket and the BamA lateral gate:

(Lines 458-471) "The β-signal motif (ζxGxx[Ω/Φ]x[Ω/Φ]) is an eight-residue consensus, and internal signal motif is composed of a nine-residue consensus. Recent structures have shown the lateral gate of BamA interacts with a 7-residue span of substrate OMPs. Interestingly, inhibitory compounds, such as darobactin, mimic only three resides of the C-terminal side of β-signal motif. Cross-linking presented here in our study showed that BamD residues R49 and G65 cross-linked to the positions 0 and 6 of the internal signal in OmpC (Fig. 6D). Both signals are larger than the assembly machineries signal binding pocket, implying that the signal might sit beyond the bounds of the signal binding pocket in BamD and the lateral gate in BamA. These finding are consistent with similar observations in other signal sequence recognition events, such as the mitochondrial targeting presequence signal that is longer than the receptor groove formed by the Tom20, the subunit of the translocator of outer membrane (TOM) complex (Yamamoto et al., 2011). The presequence has been shown to bind to Tom20 in several different conformations within the receptor groove (Nyirenda et al., 2013)."

Moreover, the distance between amino acids of BamD which cross-linked to the internal signal, R49 and Y62, is approximately 25 Å (pdbID used 7TT3). The distance of the maximum amino acid length of the internal signal of OmpC, from F280 to Y288, is approximately 22 Å (pdbID used 2J1N). This would allow for the signal to fit within the confines of the TRP motif of BamD.

**Author response image 1. sa2fig1:** 

1. The authors highlight that the sequence motifs are common among the inhibiting peptides, but do not test if this is a necessary motif to mediate the interactions. It would have been good to see if a library of non-OMP related peptides that match this motif could also inhibit or not.

With respect, this additional work would not address any biological question relevant to the function of BamD. To randomize sequences and then classify those that do or don’t fit the motif would help in refining the parameters of the β-signal motif, but that was not our intent.

We have identified the peptides from within the total sequence of an OMP, shown which peptides inhibit in an assembly assay, and then observed that the inhibitory peptides conform to a previously published (β-signal) motif.

1. In the studies that disrupt the motifs by mutagenesis, an effect was observed and attributed to disruption of the interaction of the 'internal signal'. However, the literature is filled with point mutations in OMPs that disrupt biogenesis, particular those within the membrane region. F280, Y286, V359, and Y365 are all residues that are in the membrane region that point into the membrane. Therefore, more work is needed to confirm that these mutations are in parts of a recognition motif rather than on the residues that are disrupting stability/assembly into the membrane.

As the reviewer pointed out, the side chains of the amino acids constituting the signal elements we determined were all facing the lipid side, of which Y286 and Y365 were important for folding as well as to be recognized. However, F280A and V359A had no effect on folding, but only on assembly through the BAM complex. The fact that position 0 functions as a signal has been demonstrated by peptidomimetics (Fig. 1) and point mutant analysis (Fig. 2). We appreciate this clarification and have modified the text to acknowledge that the all of the signal element faces the lipid side, which contributes to their stability in the membrane finally, and before that the BAM complex actively recognizes them and determines their orientation:

(Lines 519-526) After OMP assembly, all elements of the internal signal are positioned such that they face into the lipid-phase of the membrane. This observation may be a coincidence, or may be utilized by the BAM complex to register and orientate the lipid facing amino acids in the assembling OMP away from the formative lumen of the OMP. Amino acids at position 6, such as Y286 in OmpC, are not only component of the internal signal for binding by the BAM complex, but also act in structural capacity to register the aromatic girdle for optimal stability of the OMP in the membrane.

1. The title of Figure 3 indicates that disrupting the internal signal motif disrupts OMP assembly, however, the point mutations did not seem to have any effect. Only when both 280 and 286 were mutated was an effect observed. And even then, the trimer appeared to form just fine, albeit at reduced levels, indicating assembly is just fine, rather the rate of biogenesis is being affected.

We appreciate this point and have revised the title of Figure 3 to be:

(Lines 1070-1071) "Modifications in the putative internal signal slow the rate of OMP assembly in vivo."

1. In Figure 4, the authors attempt to quantify their blots. However, this seems to be a difficult task given the lack of quality of the blots and the spread of the intended signals, particularly of the 'int' bands. However, the more disturbing trend is the obvious reduction in signal from the post-urea treatment, even for the WT samples. The authors are using urea washes to indicate removal of only stalled substrates. However a reduction of signal is also observed for the WT. The authors should quantify this blot as well, but it is clear visually that both WT and the mutant have obvious reductions in the observable signals. Further, this data seems to conflict with Fig 3D where no noticeable difference in OmpC assembly was observed between WT and Y286A, why is this the case?

We have addressed this point by adding a statistical analysis on Fig. 4A. As the reviewer points out, BN-PAGE band quantification is a difficult task given the broad spread of the bands on these gels. Statistical analysis showed that the increase in intermediates (int) was statistically significant for Y286A at all times until 80 min, when the intermediate form signals decrease.

(Lines 1093-1096) "Statistical significance was indicated by the following: N.S. (not significant), p<0.05; *, p<0.005; **. Exact p values of intermediate formed by Wt vs Y286A at each timepoint were as follows; 20 minutes: p = 0.03077, 40 minutes: p = 0.02402, 60 minutes: p = 0.00181, 80 minutes: p = 0.0545."

Further regarding the Int. band, we correct the statement as follows.

(Lines 253-254) "Consistent with this, the assembly intermediate which was prominently observed at the OmpC(Y286A) can be extracted from the membranes with urea;"

OMP assembly in vivo has additional periplasmic chaperones and factors present in order to support the assembly process. Therefore, it is likely that some proteins were assembled properly in vivo compared to their in vitro counterparts. Such a decrease has been observed not only in *E. coli* but also in mitochondrial OMP import (Yamano et al., 2010).

1. The pull-down assays with BamA and BamD should include a no protein control at the least to confirm there is no non-specific binding to the resin. Also, no detergent was mentioned as part of the pull downs that contained BamA or OmpC, nor was it detailed if OmpC was urea solubilized.

We have performed pull down experiments with a no-protein (Ni-NTA only) control as noted (Author response image 1). The results showed that the amount of OmpC carrying through on beads only was significantly lower than the amount of OmpC bound in the presence of BamD or BamA. The added OmpC was not treated with urea, but was synthesized by in vitro translation; the in vitro translated OmpC is the standard substrate in the EMM assembly assay (Supp Fig. S1) where it is recognized by the BAM complex. Thus, we used it for pull-down as well and, to make this clearer, we have revised as follows:

**Author response image 2. sa2fig2:** Pull down assay of radio-labelled OmpC with indicated protein or Ni-NTA alone (Ni-NTA) . T; total, FT; Flow throw, W; wash, E; Elute.

(Lines 252-265) "Three subunits of the BAM complex have been previously shown to interact with the substrates: BamA, BamB, and BamD (Hagan et al., 2013; Harrison, 1996; Ieva et al., 2011). In vitro pull-down assay showed that while BamA and BamD can independently bind to the in vitro translated OmpC polypeptide (Fig .S9A), BamB did not (Fig. S9B)."

1.• The neutron reflectometry experiments are not convincing primarily due to the lack controls to confirm a consistent uniform bilayer is being formed and even if so, uniform orientations of the BamA molecules across the surface.• Further, no controls were performed with BamD alone, or with OmpC alone, and it is hard to understand how the method can discriminate between an actual BamA/BamD complex versus BamA and BamD individually being located at the membrane surface without forming an actual complex.• Previous studies have reported difficulty in preparing a complex with BamA and BamD from purified components.• Additionally, little signal differences were observed for the addition of OmpC. However, an elongated unfolded polypeptide that is nearly 400 residues long would be expected to produce a large distinct signal given that only the C-terminal portion is supposedly anchored to BAM, while the rest would be extended out above the surface.• The depiction in Figure 5D is quite misleading when viewing the full structures on the same scales with one another.

We have addressed these five points individually as follows.

i. The uniform orientation of BamA on the surface is guaranteed by the fixation through a His-tag engineered into extracellular loop 6 of BamA and has been validated in previous studies as cited in the text. Moreover, to explain this, we reconstructed another theoretical model for BamA not oriented well in the system as below. However, we found that the solid lines (after fitting) didn’t align well with the experimental data. We therefore assumed that BamA has oriented well in the membrane bilayer.

**Author response image 3. sa2fig3:** Experimental (symbols) and fitted (curves) NR profiles of BamA not oriented well in the POPC bilayer in D2O (black), GMW (blue) and H2O (red) buffer.

ii. There would be no means by which to do a control with OmpC alone or BamD alone as neither protein binds to the lipid layer chip. OmpC is diluted from urea and then the unbound OmpC is washed from the chip before NR measurements. BamD does not have an acyl group to anchor it to the lipid layer, without BamA to anchor to, it too is washed from the chip before NR measurements. We have reconstructed another theoretical model for both of BamA + BamD embedding in the membrane bilayer, and the fits were shown below. Apparently, the fits didn’t align well with the experimental data, which discriminate the BamA/BamD individually being located at the membrane surface without forming an actual complex.

**Author response image 4. sa2fig4:** Experimental (symbols) and fitted (curves) NR profiles of BamA+D embedding together in the POPC bilayer in D2O (black), GMW (blue) and H2O (red) buffer.

iii. The previous studies that reported difficulty in preparing a complex with BamA and BamD from purified components were assays done in aqueous solution including detergent solubilized BamA, or with BamA POTRA domains only. Our assay is superior in that it reports the binding of BamD to a purified BamA that has been reconstituted in a lipid bilayer.

iv. The relatively small signal differences observed for the addition of OmpC are expected, since OmpC is an elongated, unfolded polypeptide of nearly 400 residues long which, in the context of this assay, can occupy a huge variation in the positions at which it will sit with only the C-terminal portion anchored to BAM, and the rest moving randomly about and extended from the surface.

v. We appreciate the point raised and have now added a note in the Figure legend that these are depictions of the results and not a scale drawing of the structures.

1. In the crosslinking studies, the authors show 17 crosslinking sites (43% of all tested) on BamD crosslinked with OmpC. Given that the authors are presenting specific interactions between the two proteins, this is worrisome as the crosslinks were found across the entire surface of BamD. How do the authors explain this? Are all these specific or non-specific?

The crosslinking experiment using purified BamD was an effective assay for comprehensive analysis of the interaction sites between BamD and the substrate. However, as the reviewer pointed out, cross-linking was observed even at the sites that, in the context of the BAM complex, interact with BamC as a protein-protein interaction and would not be available for substrate protein-protein interactions. To complement this, analysis and to address this issue, we also performed the experiment in Fig. 6C.

In Fig. 6C, the interaction of BamD with the substrate is examined in vivo, and the results demonstrate that if BPA is introduced into the site, we designated as the substrate recognition site, it is cross-linked to the substrate. On the other hand, position 114 was found to crosslink with the substrate in vitro crosslinking, but not in vivo. It should be noted that position 114 has also been confirmed to form cross-link products with BamC, we believe that BamD-substrate interactions in the native state have been investigated. To explain the above, we have added the following description to the Results section.

(Lines 319-321) "Structurally, these amino acids locate both the lumen side of funnel-like structure (e.g. 49 or 62) and outside of funnel-like structure such as BamC binding site (e.g. 114) (fig. S12C).(Lines 350-357) Positions 49, 53, 65, and 196 of BamD face the interior of the funnel-like structure of the periplasmic domain of the BAM complex, while position 114 is located outside of the funnel-like structure (Bakelar et al., 2016; Gu et al., 2016; Iadanza et al., 2016). We note that while position 114 was cross-linked with OmpC in vitro using purified BamD, that this was not seen with in vivo cross-linking. Instead, in the context of the BAM complex, position 114 of BamD binds to the BamC subunit and would not be available for substrate binding in vivo (Bakelar et al., 2016; Gu et al., 2016; Iadanza et al., 2016)."

1. The study in Figure 6 focuses on defined regions within the OmpC sequence, but a more broad range is necessary to demonstrate specificity to these regions vs binding to other regions of the sequence as well. If the authors wish to demonstrate a specific interaction to this motif, they need to show no binding to other regions.

The region of affinity for the BAM complex was determined by peptidomimetic analysis, and the signal region was further identified by mutational analysis of OmpC. Subsequently, the subunit that recognizes the signal region was identified as BamD. In other words, in the process leading up to Fig. 6, we were able to analyze in detail that other regions were not the target of the study. We have revised the text to make clear that we focus on the signal region including the internal signal, and have not also analyzed other parts of the signal region:

(Lines 329-332) "As our peptidomimetic screen identified conserved features in the internal signal, and cross-linking highlighted the N-terminal and C-terminal TPR motifs of BamD as regions of interaction with OmpC, we focused on amino acids specifically within the β-signals of OmpC and regions of BamD which interact with β-signal."

1. The levels of the crosslinks are barely detectable via western blot analysis. If the interactions between the two surfaces are required, why are the levels for most of the blots so low?

These are western blots of cross-linked products – the efficiency of cross-linking is far less than 100% of the interacting protein species present in a binding assay and this explains why the levels for the blots are ‘so low’. We have added a sentence to the revised manuscript to make this clear for readers who are not molecular biologists:

(Lines 345-348) "These western blots reveal cross-linked products representing the interacting protein species. Photo cross-linking of unnatural amino acid is not a 100% efficient process, so the level of cross-linked products is only a small proportion of the molecules interacting in the assays."

1.• Figure 7 indicates that two regions of BamD promote OMP orientation and assembly, however, none of the experiments appears to measure OMP orientation?• Also, one common observation from panel F was that not only was the trimer reduced, but also the monomer. But even then, still a percentage of the trimer is formed, not a complete loss.

(i) We appreciate this point and have revised the title of Figure 7 to be:

(Lines 1137-1138) "Key residues in two structurally distinct regions of BamD promote β-strand formation and OMP assembly."

(ii) In our description of Fig. 7F (Lines 356-360) we do not distinguish between the amount of monomer and trimer forms, since both are reflective of the overall assembly rate i.e. assembly efficiency. Rather, we state that:

"The EMM assembly assay showed that the internal signal binding site was as important as the β-signal binding site to the overall assembly rates observed for OmpC (Fig. 7F), OmpF (fig. S15D), and LamB (fig. S15E). These results suggest that recognition of both the C-terminal β-signal and the internal signal by BamD is important for efficient protein assembly."

1.• The experiment in Fig 7B would be more conclusive if it was repeated with both the Y62A and R197A mutants and a double mutant. These controls would also help resolve any effect from crowding that may also promote the crosslinks.• Further, the mutation of R197 is an odd choice given that this residue has been studied previously and was found to mediate a salt bridge with BamA. How was this resolved by the authors in choosing this site since it was not one of the original crosslinking sites?

As stated in the text, the purpose of the experiment in Figure 7B is to measure the impact of pre-forming a β-strand in the substrate (OmpC) before providing it to the receptor (BamD). We thank the reviewer for the comment on the R197 position of BamD. The C-terminal domain of BamD has been suggested to mediate the BamA-BamD interface, specifically BamD R197 amino acid creates a salt-bridge with BamA E373 (Ricci et al., 2012). It had been postulated that the formation of this salt-bridge is not strictly structural, with R197 highlighted as a key amino acid in BamD activity and this salt-bridge acts as a “check-point” in BAM complex activity (Ricci et al., 2012, Storek et al., 2023). Our results agree with this, showing that the C-terminus of BamD acts in substrate recognition and alignment of the β-signal (Fig. 6, Fig S12). We show that amino acids in the vicinity of R197 (N196, G200, D204) cross-linked well to substrate and mutations to the β-signal prevent this interaction (Fig S12B, D). For mutational analysis of BamD, we looked then at the conservation of the C-terminus of BamD and determined R197 was the most highly conserved amino acid (Fig 6C). In order to account for this, we have adjusted the manuscript:

(Lines 376-377) "R197 has previously been isolated as a suppressor mutation of a BamA temperature sensitive strain (Ricci et al., 2012)."

(Lines 495-496) "This adds an additional role of the C-terminus of BamD beyond a complex stability role (Ricci et al., 2012; Storek et al., 2023)."

1. As demonstrated by the authors in Fig 8, the mutations in BamD lead to reduction in OMP levels for more than just OmpC and issues with the membrane are clearly observable with Y62A, although not with R197A in the presence of VCN. The authors should also test with rifampicin which is smaller and would monitor even more subtle issues with the membrane. Oddly, no growth was observed for the Vec control in the lower concentration of VCN, but was near WT levels for 3 times VCN, how is this explained?

While it would be interesting to correlate the extent of differences to the molecular size of different antibiotics such as rifampicin, such correlations are not the intended aim of our study. Vancomycin (VCN) is a standard measure of outer membrane integrity in our field, hence its use in our tests for membrane integrity.

We apologize to the reviewer as Figure 8 D-G may have been misleading. Figure 8D,E are using bamD shut-down cells expressing plasmid-borne BamD mutants. Whereas Figure 8F, G are the same strain as used in Figure 3. We have adjusted the figure as well as the figure legend:(Lines 1165-1169) D, E E coli bamD depletion cells expressing mutations at residues, Y62A and R197A, in the β-signal recognition regions of BamD were grown with of VCN. F, G, E coli cells expressing mutations to OmpC internal signal, as shown in Fig 3, grown in the presence of VCN. Mutations to two key residues of the internal signal were sensitive to the presence of VCN.

1. While Fig 8I indeed shows diminished levels for FY as stated, little difference was observed for the trimer for the other mutants compared to WT, although differences were observed for the dimer. Interestingly, the VY mutant has nearly WT levels of dimer. What do the authors postulate is going on here with the dimer to trimer transition? How do the levels of monomer compare, which is not shown?

The BN-PAGE gel system cannot resolve protein species that migrate below ~50kDa and the monomer species of the OMPs is below this size. We can’t comment on effects on the monomer because it is not visualized. The non-cropped gel image is shown here. Recently, Hussain et al., has shown that in vitro proteo-liposome system OmpC assembly progresses from a “short-lived dimeric” form before the final process of trimerization (Hussain et al., 2021). However, their findings suggest that LPS plays the final role in stimulation of dimer-to-trimer, a step well past the recognition step of the β-signals. Mutations to the internal signal of OmpC results in the formation of an intermediate, the substrate stalled on the BAM complex. This stalling, presumably, causes a hinderance to the BAM complex resulting in reduced timer and loss of dimer OmpF signal in the EMM of cells expressing OmpC double mutant strain, FY. cannot resolve protein species that migrate below ~50kDa and the monomer species of the OMPs is below this size. We can’t comment on effects on the monomer because it is not visualized. The non-cropped gel image is shown here. We have noted this in the revised text:

**Author response image 5. sa2fig5:** Non-cropped gel of Fig. 8I. the asterisk indicates a band observed in the sample loading wells at the top of the gel.

(Lines 417-418) "The dimeric form of endogenous OmpF was prominently observed in both the OmpC(WT) as well as the OmpC(VY) double mutant cells."

1. In the discussion, the authors indicate they have '...defined an internal signal for OMP assembly', however, their study is limited and only investigates a specific region of OmpC. More is needed to definitively say this for even OmpC, and even more so to indicate this is a general feature for all OMPs.

We acknowledge the reviewer's comment on this point and have expanded the statement to make sure that the conclusion is justified with the specific evidence that is shown in the paper and the supplementary data. We now state:

(Lines 444-447) "This internal signal corresponds to the -5 strand in OmpC and is recognized by BamD. Sequence analysis shows that similar sequence signatures are present in other OMPs (Figs. S5, S6 and S7). These sequences were investigated in two further OMPs: OmpF and LamB (Fig. 2C and D)."

Note, we did not state that this is a general feature for all OMPs. That would not be a reasonable proposition.

1.• In the proposed model in Fig 9, it is hard to conceive how 5 strands will form along BamD given the limited surface area and tight space beneath BAM.• More concerning is that the two proposal interaction sites on BamD, Y62 and R197, are on opposite sides of the BamD structure, not along the same interface, which makes this model even more unlikely.• As evidence against this model, in Figure 9E, the two indicates sites of BamD are not even in close proximity of the modeled substrate strands.

We can address the reviewer’s three concerns here:

i. The first point is that the region (formed by BamD engaged with POTRA domains 1-2 and 5 of BamA) is not sufficient to accommodate five β-strands. Structural analysis reveals that the interaction between the N-terminal side of BamD and POTRA1-2 is substantially changed the conformation by substrate binding, and that this surface is greatly extended. This surface does have enough space to accommodate five beta-strands, as now documented in Fig. 9D, 9E using the latest structures (7TT5 and 7TT2) as illustrations of this. The text now reads:

(Lines 506-515) "Spatially, this indicates the BamD can serve to organize two distinct parts of the nascent OMP substrate at the periplasmic face of the BAM complex, either prior to or in concert with, engagement to the lateral gate of BamA. Assessing this structurally showed the N-terminal region of BamD (interacting with the POTRA1-2 region of BamA) and the C-terminal region of BamD (interacting with POTRA5 proximal to the lateral gate of BamA) (Bakelar et al., 2016; Gu et al., 2016; Tomasek et al., 2020) has the N-terminal region of BamD changing conformation depending on the folding states of the last four β-strands of the substrate OMP, EspP (Doyle et al., 2022). The overall effect of this being a change in the dimensions of this cavity change, a change which is dependent on the folded state of the substrate engaged in it (Fig 9 B-E)."

ii. The second point raised regards the orientation of the substrate recognition residues of BamD. Both Y62A and R197 were located on the lumen side of the funnel in the EspP-BAM transport intermediate structure (PDBID;7TTC); Y62A is relatively located on the edge of BamD, but given that POTRA1-2 undergoes a conformational change and opens this region, as described above, both are located in locations where they could bind to substrates. This was explained in the following text in the results section of revised manuscript.

(Lines 377-379) "Each residue was located on the lumen side of the funnel-like structure in the EspP-BAM assembly intermediate structure (PDBID; 7TTC) (Doyle et al., 2022)."

**Reviewer #2 (Public Review):"Previously, using bioinformatics study, authors have identified potential sequence motifs that are common to a large subset of beta-barrel outer membrane proteins in gram negative bacteria. Interestingly, in that study, some of those motifs are located in the internal strands of barrels (not near the termini), in addition to the well-known "beta-signal" motif in the C-terminal region.Here, the authors carried out rigorous biochemical, biophysical, and genetic studies to prove that the newly identified internal motifs are critical to the assembly of outer membrane proteins and the interaction with the BAM complex. The author's approaches are rigorous and comprehensive, whose results reasonably well support the conclusions. While overall enthusiastic, I have some scientific concerns with the rationale of the neutron refractory study, and the distinction between "the intrinsic impairment of the barrel" vs "the impairment of interaction with BAM" that the internal signal may play a role in. I hope that the authors will be able to address this.Strengths:1. It is impressive that the authors took multi-faceted approaches using the assays on reconstituted, cell-based, and population-level (growth) systems.1. Assessing the role of the internal motifs in the assembly of model OMPs in the absence and presence of BAM machinery was a nice approach for a precise definition of the role.Weaknesses:1. The result section employing the neutron refractory (NR) needs to be clarified and strengthened in the main text (from line 226). In the current form, the NR result seems not so convincing.What is the rationale of the approach using NR?

We have now modified the text to make clear that:

(Lines 276-280) "The rationale to these experiments is that NR provides: (i) information on the distance of specified subunits of a protein complex away from the atomically flat gold surface to which the complex is attached, and (ii) allows the addition of samples between measurements, so that multi-step changes can be made to, for example, detect changes in domain conformation in response to the addition of a substrate."

What is the molecular event (readout) that the method detects?

We have now modified the text to make clear that:

(Lines 270-274) "While the biochemical assay demonstrated that the OmpC(Y286A) mutant forms a stalled intermediate with the BAM complex, in a state in which membrane insertion was not completed, biochemical assays such as this cannot elucidate where on BamA-BamD this OmpC(Y286A) substrate is stalled."

What are "R"-y axis and "Q"-x axis and their physical meanings (Fig. 5b)?

The neutron reflectivity, R, refers to the ratio of the incoming and exiting neutron beams and it is measured as a function of Momentum transfer Q, which is defined as Q=4π sinθ/λ, where θ is the angle of incident and λ is the neutron wavelength. R(Q)is approximately given byR(Q)=16π2/ Q2 |ρ(Q)|2, where R(Q) is the one-dimensional Fourier transform of ρ(z), the scattering length density (SLD) distribution normal to the surface. SLD is the sum of the coherent neutron scattering lengths of all atoms in the sample layer divided by the volume of the layer. Therefore, the intensity of the reflected beams is highly dependent on the thickness, densities and interface roughness of the samples. This was explained in the following text in the method section of revised manuscript.

(Lines 669-678) "Neutron reflectivity, denoted as R, is the ratio of the incoming to the exiting neutron beams. It’s calculated based on the Momentum transfer Q, which is defined by the formula Q=4π sinθ/λ, where θ represents the angle of incidence and λ stands for the neutron wavelength. The approximate value of R(Q) can be expressed as R(Q)=16π2/ Q2 |ρ(Q)|2, where R(Q) is the one-dimensional Fourier transform of ρ(z), which is the scattering length density (SLD) distribution perpendicular to the surface. SLD is calculated by dividing the sum of the coherent neutron scattering lengths of all atoms in a sample layer by the volume of that layer. Consequently, factors such as thickness, volume fraction, and interface roughness of the samples significantly influence the intensity of the reflected beams."

How are the "layers" defined from the plot (Fig. 5b)?

The “layers” in the plot (Fig. 5b) represent different regions of the sample being studied. In this study, we used a seven-layer model to fit the experimental data chromium - gold - NTA - HIS8 - β-barrel - P3-5 - P1-2. This was explained in the following text in the figure legend of revised manuscript.(Lines 1115-1116) The experimental data was fitted using a seven-layer model: chromium - gold - NTA - His8 - β-barrel - P3-5 - P1-2.

What are the meanings of "thickness" and "roughness" (Fig. 5c)?

We used neutron reflectometry to determine the relative positions of BAM subunits in a membrane environment. The binding of certain subunits induced conformational changes in other parts of the complex. When a substrate membrane protein is added, the periplasmic POTRA domain of BamA extends further away from the membrane surface. This could result in an increase in thickness as observed in neutron reflectometry measurements.

As for roughness, it is related to the interface properties of the sample. In neutron reflectometry, the intensity of the reflected beams is highly dependent on the thickness, densities, and interface roughness of the samples. An increase in roughness could suggest changes in these properties, possibly due to protein-membrane interactions or structural changes within the membrane.

(Lines 1116-1120) "Table summarizes of the thickness, roughness and volume fraction data of each layer from the NR analysis. The thickness refers to the depth of layered structures being studied as measured in Å. The roughness refers to the irregularities in the surface of the layered structures being studied as measured in Å."

What does "SLD" stand for?

We apologize for not explaining abbreviation when the SLD first came out. We explained it in revised manuscript. (Line 298)

1. In the result section, "The internal signal is necessary for insertion step of assembly into OM"This section presents an important result that the internal beta-signal is critical to the intrinsic propensity of barrel formation, distinct from the recognition by BAM complex. However, this point is not elaborated in this section. For example, what is the role of these critical residues in the barrel structure formation? That is, are they involved in any special tertiary contacts in the structure or in membrane anchoring of the nascent polypeptide chains?

We appreciate the reviewer's comment on this point. Both position 0 and position 6 appear to be important amino acids for recognition by the BAM complex, since mutations introduced at these positions in peptide 18 prevent competitive inhibition activity.

In terms of the tertiary structure of OmpC, position 6 is an amino acid that contributes to the aromatic girdle, and since Y286A and Y365A affected OMP folding as measured in folding experiments, it is perhaps their position in the aromatic girdle that contributes to the efficiency of β-barrel folding in addition to its function as a recognition signal. We have added a sentence in the revised manuscript:

(Lines 233-236) "Position 6 is an amino acid that contributes to the aromatic girdle. Since Y286A and Y365A affected OMP folding as measured in folding experiments, their positioning into the aromatic girdle may contributes to the efficiency of β-barrel folding, in addition to contributing to the internal signal."

The mutations made at position 0 had no effect on folding, so this residue may function solely in the signal. Given the register of each β-strand in the final barrel, the position 0 residues have side-chains that face out into the lipid environment. From examination of the OmpC crystal structure, the residue at position 0 makes no special tertiary contacts with other, neighbouring residues.

**Reviewer #1 (Recommendations For The Authors):**
Minor critiques (in no particular order):1. Peptide 18 was identified based on its strong inhibition for EspP assembly but another peptide, peptide 23, also shows inhibition and has no particular consensus.

We would correct this point. Peptide 23 has a strong consensus to the canonical β-signal. We had explained the sequence consensus of β-signal in the Results section of the text. In the third paragraph, we have added a sentence indicating the relationship between peptide 18 and peptide 23.

(Lines 152-168) "Six peptides (4, 10, 17, 18, 21, and 23) were found to inhibit EspP assembly (Fig. 1A). Of these, peptide 23 corresponds to the canonical β-signal of OMPs: it is the final β-strand of OmpC and it contains the consensus motif of the β-signal (ζxGxx[Ω/Φ]x[Ω/Φ]). The inhibition seen with peptide 23 indicated that our peptidomimetics screening system using EspP can detect signals recognized by the BAM complex. In addition to inhibiting EspP assembly, five of the most potent peptides (4, 17, 18, 21, and 23) inhibited additional model OMPs; the porins OmpC and OmpF, the peptidoglycan-binding OmpA, and the maltoporin LamB (fig. S3). Comparing the sequences of these inhibitory peptides suggested the presence of a sub-motif from within the β-signal, namely [Ω/Φ]x[Ω/Φ] (Fig. 1B). The sequence codes refer to conserved residues such that: ζ, is any polar residue; G is a glycine residue; Ω is any aromatic residue; Φ is any hydrophobic residue and x is any residue (Hagan et al., 2015; Kutik et al., 2008). The non-inhibitory peptide 9 contained some elements of the β-signal but did not show inhibition of EspP assembly (Fig. 1A).

Peptide 18 also showed a strong sequence similarity to the consensus motif of the β-signal (Fig. 1B) and, like peptide 23, had a strong inhibitory action on EspP assembly (Fig. 1A). Variant peptides based on the peptide 18 sequence were constructed and tested in the EMM assembly assay (Fig. 1C)."

1. It is unclear why the authors immediately focused on BamD rather than BamB, given that both were mentioned to mediate interaction with substrate. Was BamB also tested?

We thank the reviewer for this comment. Following the reviewer's suggestion, we have now performed a pull-down experiment on BamB and added it to Fig. S9. We also modified the text of the results as follows.

(Lines 262-265) "Three subunits of the BAM complex have been previously shown to interact with the substrates: BamA, BamB, and BamD (Hagan et al., 2013; Harrison, 1996; Ieva et al., 2011). In vitro pull-down assay showed that while BamA and BamD can independently bind to the in vitro translated OmpC polypeptide (Fig .S9A), BamB did not (Fig. S9B)."

1. For the in vitro folding assays of the OmpC substrates, labeled and unlabeled, no mention of adding SurA or any other chaperone which is known to be important for mediating OMP biogenesis in vitro.

We appreciate the reviewer’s concerns on this point, however chaperones such as SurA are non-essential factors in the OMP assembly reaction mediated by the BAM complex: the surA gene is not essential and the assembly of OMPs can be measured in the absence of exogenously added SurA.It remains possible that addition of SurA to some of these assays could be useful in detailing aspects of chaperone function in the context of the BAM complex, but that was not the intent of this study.

1. For the supplementary document, it would be much easier for the reader to have the legends groups with the figures.

Following the reviewer's suggestion, we have placed the legends of Supplemental Figures together with each Figure.

1. Some of the figures and their captions are not grouped properly and are separated which makes it hard to interpret the figures efficiently.

We thank the reviewer for this comment, we have revised the manuscript and figures to properly group the figures and captions together on a single page.

1. The authors begin their 'Discussion' with a question (line 454), however, they don't appear to answer or even attempt to address it; suggest removing rhetorical questions.

As per the reviewers’ suggestion, we removed this question.

1. Line 464, 'unbiased' should be removed. This would imply that if not stated, experiments are 'negatively' biased.

We removed this word and revised the sentence as follows:

(Lines 431-433) "In our experimental approach to assess for inhibitory peptides, specific segments of the major porin substrate OmpC were shown to interact with the BAM complex as peptidomimetic inhibitors."

1. Lines 466-467; '...go well beyond expected outcomes.' What does this statement mean?

Our peptidomimetics led to unexpected results in elucidating the additional essential signal elements. The manuscript was revised as follows:

(Lines 433-435) "Results for this experimental approach went beyond expected outcomes by identifying the essential elements of the signal Φxxxxxx[Ω/Φ]x[Ω/Φ] in β-strands other than the C-terminal strand."

1. Line 478; '...rich information that must be oversimplified...'?

We appreciate the reviewer’s pointed out. For more clarity, the manuscript was revised as follows:

(Lines 450-453) "The abundance of information which arises from modeling approaches and from the multitude of candidate OMPs, is generally oversimplified when written as a primary structure description typical of the β-signal for bacterial OMPs (i.e. ζxGxx[Ω/Φ]x[Ω/Φ]) (Kutik et al., 2008)."

1. There are typos in the supplementary figures.

We have revised and corrected the Supplemental Figure legends.

**Reviewer #2 (Recommendations For The Authors):**
1. In Supplementary Information, I recommend adding the figure legends directly to the corresponding figures. Currently, it is very inconvenient to go back and forth between legends and figures.

Following the reviewer's suggestion, we have placed the legends of Supplemental Figures together with each Figure.

1. Line 94 (p.3): "later"Lateral?

Yes. We have corrected this.

1. Line 113 (p.3): The result section, "Peptidomimetics derived from *E. coli* OmpC inhibit OMP assembly" Rationale of the peptide inhibition assay is not clear. How can the peptide sequence that effectively inhibit the assembly interpreted as the b-assembly signal? By competitive binding to BAM or by something else? What is the authors' hypothesis in doing this assay?

In revision, we have added following sentence to explain the aim and design of the peptidomimetics:

(Lines 140-145) "The addition of peptides with BAM complex affinity, such as the OMP β-signal, are capable of exerting an inhibitory effect by competing for binding of substrate OMPs to the BAM complex (Hagan et al., 2015). Thus, the addition of peptides derived from the entirety of OMPs to the EMM assembly assay, which can evaluate assembly efficiency with high accuracy, expects to identify novel regions that have affinity for the BAM complex."

1. Line 113- (p.3) and Fig. S1: The result section, "Peptidomimetics derived from *E. coli* OmpC inhibit OMP assembly"Some explanation seems to be needed why b-barrel domain of EspP appears even without ProK?

We appreciate the reviewer’s pointed out. We added following sentence to explain:

(Lines 128-137) "EspP, a model OMP substrate, belongs to autotransporter family of proteins. Autotransporters have two domains; (1) a β-barrel domain, assembled into the outer membrane via the BAM complex, and (2) a passenger domain, which traverses the outer membrane via the lumen of the β-barrel domain itself and is subsequently cleaved by the correctly assembled β-barrel domain (Celik et al., 2012). When EspP is correctly assembled into outer membrane, a visible decrease in the molecular mass of the protein is observed due to the self-proteolysis. Once the barrel domain is assembled into the membrane it becomes protease-resistant, with residual unassembled and passenger domains degraded (Leyton et al., 2014; Roman-Hernandez et al., 2014)."

1. Line 186 (p.6): "Y285"Y285A?

We have corrected the error, it was Y285A.

1. Lines 245- (p. 7)/ Lines 330- (p. 10)It needs to be clarified that the results described in these paragraphs were obtained from the assays with EMM.

We appreciate the reviewer’s concerns on these points. For the first half, the following text was added at the beginning of the applicable paragraph to indicate that all of Fig. 4 is the result of the EMM assembly assay.

(Line 241) "We further analyzed the role of internal β-signal by the EMM assembly assay.At the second half, we used purified BamD but not EMM. We described clearly with following sentence."

(Lines 316-318) "We purified 40 different BPA variants of BamD, and then irradiated UV after incubating with 35S-labelled OmpC."